# Ethnic and region-specific genetic risk variants of stroke and its comorbid conditions can define the variations in the burden of stroke and its phenotypic traits

**Rashmi Sukumaran[1,2], Achuthsankar S Nair[2], Moinak Banerjee[1]***

[1]Human Molecular Genetics Laboratory, Rajiv Gandhi Centre for Biotechnology, Thiruvananthapuram, India; [2]Department of Computational Biology and Bioinformatics, University of Kerala, Thiruvananthapuram, India

**\*For correspondence:**
mbanerjee@rgcb.res.in

**Competing interest:** The authors declare that no competing interests exist.

**Abstract** Burden of stroke differs by region, which could be attributed to differences in comorbid conditions and ethnicity. Genomewide variation acts as a proxy marker for ethnicity, and comorbid conditions. We present an integrated approach to understand this variation by considering prevalence and mortality rates of stroke and its comorbid risk for 204 countries from 2009 to 2019, and Genome-wide association studies (GWAS) risk variant for all these conditions. Global and regional trend analysis of rates using linear regression, correlation, and proportion analysis, signifies ethno-geographic differences. Interestingly, the comorbid conditions that act as risk drivers for stroke differed by regions, with more of metabolic risk in America and Europe, in contrast to high systolic blood pressure in Asian and African regions. GWAS risk loci of stroke and its comorbid conditions indicate distinct population stratification for each of these conditions, signifying for population-specific risk. Unique and shared genetic risk variants for stroke, and its comorbid and followed up with ethnic-specific variation can help in determining regional risk drivers for stroke. Unique ethnic-specific risk variants and their distinct patterns of linkage disequilibrium further uncover the drivers for phenotypic variation. Therefore, identifying population- and comorbidity-specific risk variants might help in defining the threshold for risk, and aid in developing population-specific prevention strategies for stroke.

## eLife assessment

This paper provides a **useful** analysis of the variation of the burden of strokes across geographic regions, finding differences in the relationship between strokes and their comorbidities. This dataset and the correlations found within will be a resource for directing the focus of future investigations. The results are technically **solid**, but there are cases where statistical analyses are yet to be carried out to support statements of statistical significance.

## Introduction

Stroke affects over 101 million people worldwide and is ranked the second most fatal disease in the world, with 6.5 million deaths in 2019 (*GBD 2019 Stroke Collaborators, 2021*). Comorbid conditions of stroke are critical contributors to burden of stroke and the duration of the comorbid conditions can further determine the severity of stroke risk or mortality. Prevalence for comorbid conditions range from 43% to 94% and estimates can go as high as 99% above 66 years of age (*Gallacher et al., 2019*). Prevalence and mortality risk in stroke have often been evaluated from socio-economic viewpoint,

but it is also critical to understand the differences in drivers such as comorbid conditions. It is the accumulated risk of comorbid conditions that enhance the risk of stroke further. Are these comorbid conditions differentially impacted by socio-economic factors and ethnogeographic factors. This was clearly evident in COVID era, when COVID-19 differentially impacted the risk of stroke, possibly due to its differential influence on the comorbidities of stroke.

Mortality in stroke, its subtypes, and their comorbid conditions have a strong ethnic bias (*Tarko et al., 2022*; *Gardener et al., 2020*; *Mkoma et al., 2020*). Genetics act as surrogate markers for ethnogeographic indices. It is important to understand which comorbid conditions are influenced by socio-economic indices, and how they impact the risk of stroke and their underlying genetic basis. A Danish study reported the effect sizes of association with comorbid conditions for stroke to have 15% higher mortality risk in presence of diabetes mellitus with end-organ damage, 20% for peripheral vascular disease, 25% for chronic pulmonary disease, 35% for congestive heart failure and atrial flutter, 45% for moderate to severe renal disease, and 1.8- to 2.4-fold for mild-to-severe liver disease (*Schmidt et al., 2014*). A UK Biobank study on stroke multimorbidity reported 1.5× higher risk of mortality in those with two additional comorbidities and a ≈2.5× higher risk of mortality in those with ≥5 comorbidities over 7 years (*Gallacher et al., 2018*). Thus, the differential impact of comorbid or multimorbid conditions contributing to the additive effect of illness burden needs to be addressed from an ethnogenetic perspective. Devising an appropriate strategy for prevention of stroke burden, needs a careful evaluation of the underlying genetic signature for each of these comorbid conditions and distinguishing their ethnic bias.

The objective of the study was to understand what determines the differences in stroke burden around the globe. Variations in burden of stroke could be influenced by comorbid conditions, and incidentally both stroke and its comorbid conditions can be influenced by ethnogeographic factors and genetics can act as a stable proxy marker for all. To resolve this, we considered the prevalence and mortality of a total of 11 disease conditions, consisting of stroke and its comorbid conditions, across different continents and ethnicities from 2009 to 2019. The disease conditions were further stratified as per their ethnogeographic locations and their genetic risk variants extrapolated from GWAS data. This study would provide insights on the regional patterns of the burden of stroke and its comorbid conditions, and help in resolving it from an ethnic and genetic viewpoint. These insights would further aid in developing and strategizing regional- and ethnic-specific needs for prevention of the risk of comorbid conditions and stroke.

## Results

### Global mortality, incidence and prevalence rates of stroke and its comorbid conditions

Globally, stroke ranks as the second most fatal disease in 2019 (84.2/100,000, 95%UI 76.8–90.2) among the eight diseases analyzed, preceded by ischemic heart disease (IHD; 117.9/100,000, 95%UI 107.8–125.9) as shown in *Figure 1—figure supplement 1* and *Supplementary file 1*. High systolic blood pressure (high SBP) ranks as the most fatal comorbid disease condition with an age-standardized mortality rate (ASMR) of 138.9/100,000 [95%UI 121.3–155.7] among all conditions. Within stroke subtypes, ischemic stroke ranks highest globally with ASMR of 43.5/100,000 [95%UI 39.08–46.8], followed by intracerebral hemorrhage (ICH) at 36.0/100,000 [95%UI 32.9–38.7] and subarachnoid hemorrhage (SAH) at 4.7/100,000 [95%UI 4.1–5.2]. The ranking of the ASMRs of the diseases follows the same trend throughout the last decade with minor exceptions, where high body mass index (high BMI) improved its rank in 2014 by swapping with high low-density lipoprotein cholesterol (high LDL), and type 1 diabetes (T1D) dethroned chronic kidney disease (CKD) in 2019.

Global trends in crude and age-standardized incidence rates (ASIRs) show that stroke incidence ranks fourth, while IHD is on top followed by type 2 diabetes (T2D) and CKD (*Figure 1—figure supplement 1*, *Supplementary file 1*). Crude incidence rates of stroke and its subtypes increased in the last decade but ASIRs decreased, with exception of ischemic stroke, where an increase in ASIR was observed. Among other diseases, IHD, T2D, and T1D show a continuous increase in the last decade, with T2D (280.1/100,000, 95%UI 258.8–303.9) surpassing IHD (274.0/100,000, 95%UI 242.9–306.4) in 2019 in crude rates.

The global trend between crude and age-standardized prevalence rates (ASPRs) revealed that the ranking of stroke remains at sixth position throughout the years (*Figure 1—figure supplement 1*, *Supplementary file 1*). Among stroke subtypes, ischemic stroke ranks highest. The comorbid conditions, high SBP and high BMI, ranks first and second followed by CKD and T2D, and interestingly all rank above stroke. We also observe that though there is a continuous increase in prevalence rates in the last decade, the ranking of stroke or its comorbid conditions does not change over the years, with the exception of T1D ASPRs overtaking ICH. We were keen to understand if these global trends of ASMR and ASPR are influenced by region and ethnicity.

## Ethno-regional differences in mortality and prevalence of stroke and its major comorbid conditions

We observed interesting patterns of ASMRs of stroke, its subtypes, and its major comorbidities across different regions over the years as shown in *Figure 1a* and *Figure 1—figure supplement 2*, *Table 1* and *Supplementary file 1*. When assessed in terms of ranks, high SBP is the most fatal condition followed by IHD in all regions, except Oceania where IHD and high SBP swap ranks. Africa (206.2/100,000, 95%UI 177.4–234.2) and Middle East (198.6/100,000, 95%UI 162.8–234.4) have the highest ASMR for high SBP, even though they rank as only the third and sixth most populous continents (*Figure 1—figure supplement 3*), respectively. Both high SBP (−0.64% to −2.25% estimated annual percentage change [EAPC]) and IHD (−0.45% to −1.17% EAPC) show a decreasing trend for ASMRs in all regions. However, only Europe shows a significant decrease for high SBP (−2.26%; p = 0.009) and IHD (−2.37%; p = 0.006) in the decade. Stroke has a decreasing trend for ASMRs with East Asia and Europe showing a significant decrease of −2.2% (p = 0.021) and −2.6% (p = 0.03), respectively. Stroke has the highest mortality in East Asia (127.1/100,000, 95%UI 104.9–150.5 in 2019), and is the only region that ranks stroke higher than IHD. Though Europe, Middle East, and Central and South Asia have ASMRs similar to global rates for stroke, Central and South Asia ranks stroke as the third most fatal factor, while America, Europe, and Middle East ranks it fifth. Oceania (62.1/100,000, 95%UI 34.1–90.2) and America (40.3/100,000, 95%UI 36.2–43.1) have lowest rates for stroke in 2019.

Among the stroke subtypes, ICH and ischemic stroke show maximum ethnogenetic differences in mortality rates (14.1/100,000–61.4/100,000) and ranking (4th to 9th) in 2019. While ICH shows a significant decrease in ASMR in East Asia (−3.53%, p = 0.009), ischemic stroke shows a significant decrease in Europe (−2.37%, p = 0.06). High BMI and high LDL rank in the top 5 but their mortality rates differed across all regions, with the highest rates for both in the Middle East. Only Europe shows a significant decrease in high LDL (−2.53%, p = 0.03) over the decade. The Middle East has the highest ASMRs due to IHD and high SBP, followed by Africa, Central and South Asia, East Asia, and Europe, all having rates higher than global. All continents have similar mortality rates for T2D and CKD across the years, except Oceania, where the T2D rate is nearly three times CKD rate. Africa has the highest mortality rate for T1D (1.59/100,000, 95%UI 1.2–1.9).

ASPRs also showed an interesting pattern of distribution, and, in contrast to mortality, showed an increase over the decade (*Figure 1b* and *Figure 1—figure supplement 4*, *Table 2* and *Supplementary file 1*). Highest ASPRs were observed for high SBP across all regions, except America, Middle East, and Oceania, where high BMI has most prevalence. While EAPC of prevalence of high SBP showed significant decrease in all regions (−0.38% to −1.77%), except Central and South Asia, high BMI showed a significant increase (1.77–5.6%) in all (*Table 2*). The ASPR ranking of CKD and T2D rose to top 5, in sharp contrast to their ASMR rankings. Prevalence of CKD (EAPC 0.24–0.7%) and T2D (EAPC 0.6–2.18%) is significantly increasing in all regions. For all other diseases, the pattern of ranking and rates across regions were stable with minor exceptions. Stroke ranks sixth for ASPRs in all regions, and it is interesting to note that ASPRs of all the comorbid conditions, except T1D, rank above stroke. While Europe shows a significant decline in ASPRs of stroke (−0.9%, p = 0.008) and ischemic stroke (−0.85%, p = 0.03) across the years, East Asia shows a significant increase for stroke (0.7%, p = 0.02) and ischemic stroke (1.09%, p = <0.001). Globally, T1D swapped its prevalence ranking with ICH in 2014, largely influenced by the significant increase in prevalence in Middle East (2.83%, p = <0.001). However, the highest prevalence of T1D is in Europe and Oceania. The prevalence of IHD has remained nearly constant in all continents in the last decade, except Oceania (−0.43%), America (−0.49%), and Middle East (−0.27%) which shows a significant decrease. In 2019, Middle East has the highest prevalence for IHD (4843.02/100,000, 95%UI 4243.02–5442.58), while America

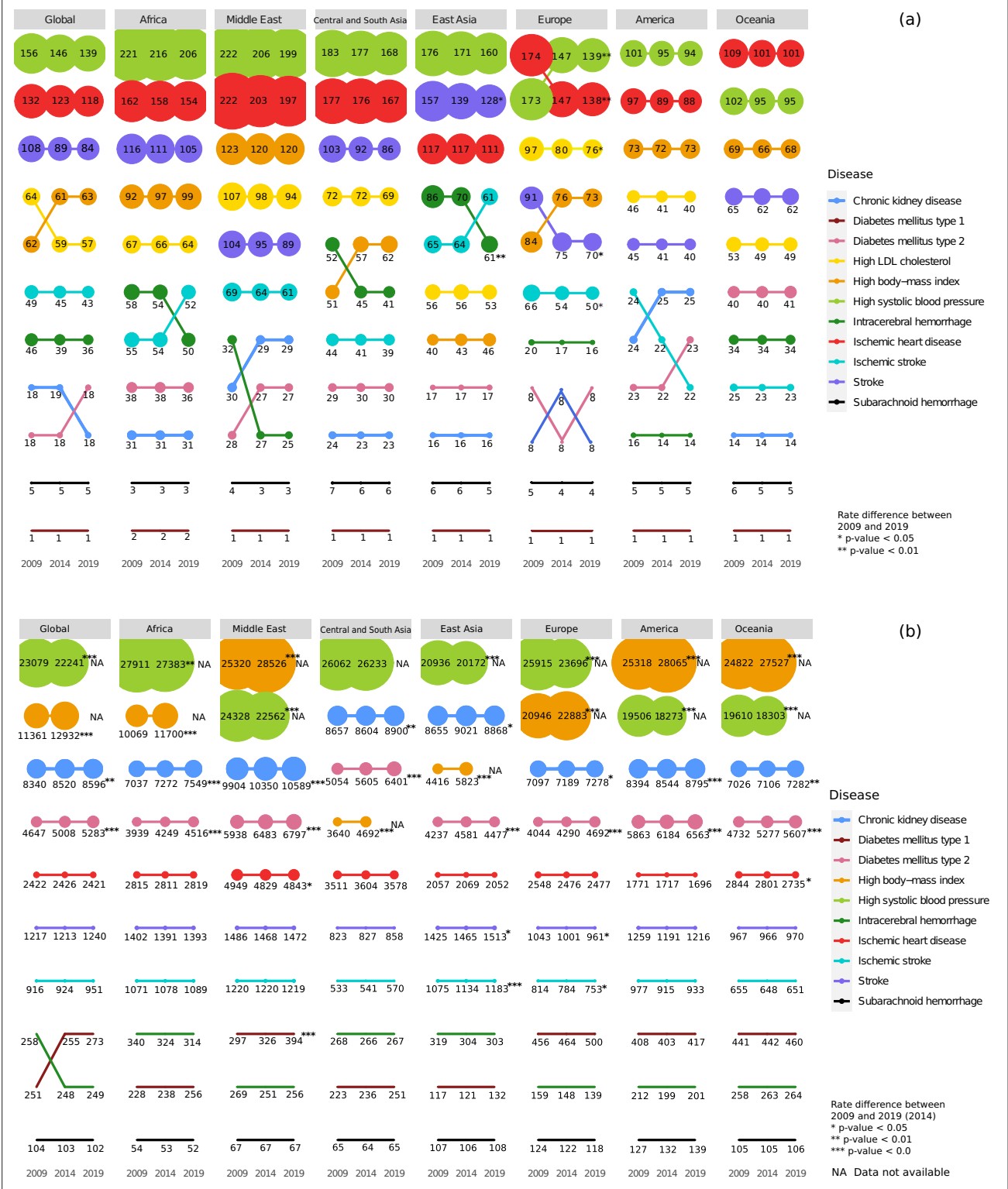

**Figure 1.** Regional (**a**) mortality rates and (**b**) prevalence rates for stroke, its subtypes, and comorbid factors. The figure shows the age-standardized mortality and prevalence rates per 100,000 people for stroke, its subtypes, and its comorbid factors in 2009, 2014, and 2019 in the geographical regions. The size of the points indicates the rate and position indicates rank.

The online version of this article includes the following figure supplement(s) for figure 1:

**Figure supplement 1.** Global incidence, mortality, and prevalence rates for stroke, its subtypes, and comorbid factors.

*Figure 1 continued on next page*

*Figure 1 continued*

**Figure supplement 2.** Boxplots of point estimates and uncertainty intervals of age-standardized mortality rates of stroke, its subtypes, and comorbid conditions across regions from 2009 to 2019.

**Figure supplement 3.** Population across different continents in 2009, 2014, and 2019.

**Figure supplement 4.** Boxplots of point estimates and uncertainty intervals of age-standardized prevalence rates of stroke, its subtypes, and comorbid conditions across regions from 2009 to 2019.

(1695.6/100,000, 95%UI 1530.8–1871.9) has the lowest ASPR, less than half the rate of the Middle East. We were further keen to understand if these regional differences in mortality and prevalence rates also reflect a socio-economic bias and if so, does it reflect in a category of comorbid conditions.

## Contribution of metabolic risk and hypertension in stroke based on ethnogenetic locations

When the prevalence and mortality rates of stroke and its comorbidities were grouped into three groups, namely strokes, metabolic disorders, and high SBP, we find that out of the three proportional mortalities shown in *Figure 2*, strokes group has the highest proportion (37.1–47.2%) across all years and regions, except Oceania and America, where instead, metabolic disorders have the highest proportion (39.1–42.2%) that is significantly higher compared to global proportion (*Supplementary file 1*). East Asia has the highest proportional mortality for strokes among all regions in all 3 years (44.7–47.2%). This was in sharp contrast to the prevalence proportion of strokes (4.6–9.1%), which was the least among the three groups, with the highest proportional prevalence for strokes being 8.9–9.1% in Central and South Asia. The proportional mortality for high SBP (22.0–30.5%) is very similar across the regions. Metabolic disorders have significantly higher proportional prevalence compared to global in Middle East, Oceania, and America, the highest being in America (63.9%, *Supplementary file 1*). Asian and African regions have lowest proportional prevalence for metabolic disorders, with Central and South Asia having significantly lower proportions. However, these regions have the highest prevalence proportion for high SBP (46.6–54.3%), with Central and South Asia having a significantly higher proportion compared to global. On the other hand, the Middle East, Oceania, and America have significantly lower proportional prevalence compared to global. We were further keen to understand the correlation among ASMR and ASPR of comorbid conditions among ethnogeographic regions.

## Correlation among prevalence and mortality rates based on ethnogeographic region

Correlation between ASMRs and ASPRs for stroke and all comorbid conditions across each ethnogeographic location is shown in *Figure 3*. High SBP prevalence and mortality show a strong positive correlation in Central and South Asia but a strong negative correlation in the Middle East. The prevalence and mortality of high BMI have a strong negative correlation in Central and South Asia and Middle East populations, but a strong positive correlation in East Asia. Prevalence of CKD has negative correlation with mortality rates in East Asia. Prevalence of T2D has negative correlation with mortality rates in Oceania and positive correlation in America. It is interesting to note that there was not much of a correlation in mortality and prevalence rates for most of the conditions. For overall stroke, though minor correlations between various ethnicities are seen, this becomes alarmingly clear in the stroke subtypes. The correlation matrix of mortality and prevalence rates of stroke and its comorbidities does reflect strong ethnogeographic distinctions, which formed the basis of further investigation on the genetic basis of stroke and its comorbidities.

## Ethnogeographic stratification of stroke and its comorbidities based on GWAS data

To resolve the ethnogeographic distinctions for stroke and its major comorbid conditions based on their genetic risk, we considered all GWAS loci for stroke and its major comorbid conditions, and subjected it to stratification analysis. From the GWAS loci, we observed a distinct population structure that distinguished ethnogeographic populations based on their genetic signatures (*Figure 4*). For all diseases, except high BMI, the individuals clustered into five groups, each corresponding with the five

**Table 1.** Estimated annual percentage change (EAPC) from 2009 to 2019 of age-standardized mortality rates (ASMRs) of stroke and its comorbid conditions.

ASMR per 100,000 for stroke and its comorbid conditions for 2009 and 2019, as well as EAPC from 2009 to 2019 is shown. 95% uncertainty interval is shown in parenthesis, statistically significant intervals are highlighted in bold.

| Region | Chronic kidney disease | | | High body mass index (BMI) | | |
| --- | --- | --- | --- | --- | --- | --- |
| | 2019 | 2009 | 2009–2019 EAPC | 2019 | 2009 | 2009–2019 EAPC |
| Global | 18.29 (19.6, 16.72) | 18.41 (19.22, 17.1) | –0.07 (–4.32, 4.37) | 62.59 (89.13, 39.92) | 61.66 (90.2, 37.44) | 0.16 (–2.2, 2.57) |
| Africa | 30.58 (27.11, 34.05) | 31.18 (28.37, 33.99) | –0.12 (–3.41, 3.27) | 98.83 (64.77, 136.6) | 91.64 (58.6, 130.04) | 0.88 (–1.02, 2.82) |
| Central and South Asia | 22.74 (17.68, 27.81) | 23.72 (18.93, 28.5) | 0.24 (–4.07, 3.73) | 62.07 (40.97, 83.17) | 51.31 (30.2, 72.47) | 2.0 (–0.5, 4.56) |
| East Asia | 15.8 (11.17, 20.42) | 16.22 (12.1, 20.34) | –0.38 (–4.9, 4.36) | 46.41 (36.68, 56.14) | 39.76 (32.9, 46.61) | 1.45 (–1.39, 4.38) |
| Europe | 8.18 (7.06, 9.3) | 8.19 (7.1, 9.29) | 0.11 (–6.17, 6.81) | 73.34 (60.31, 86.36) | 84.45 (69.2, 99.71) | –1.42 (–3.49, 0.7) |
| Middle East | 28.68 (22.76, 34.59) | 29.75 (22.4, 37.14) | –0.46 (–3.85, 3.06) | 120.14 (100.3, 139.9) | 122.6 (98.3, 146.9) | –0.23 (–1.91, 1.48) |
| Oceania | 13.71 (9.67, 17.74) | 13.6 (9.16, 18.04) | 0.17 (–4.76, 5.36) | 67.52 (41.1, 93.93) | 69.2 (42.2, 96.19) | –0.28 (–2.53, 2.02) |
| America | 24.95 (22.85, 26.81) | 23.83 (22.22, 24.68) | 0.48 (–3.24, 4.34) | 72.83 (48.62, 97.9) | 73.34 (49.02, 99.92) | –0.01 (–2.17, 2.23) |

| Region | High low-density lipoprotein (LDL) | | | High systolic blood pressure (SBP) | | |
| --- | --- | --- | --- | --- | --- | --- |
| | 2019 | 2009 | 2009–2019 EAPC | 2019 | 2009 | 2009–2019 EAPC |
| Global | 56.51 (73.62, 41.83) | 63.9 (82.9, 47.71) | –1.29 (–3.65, 1.13) | 138.88 (155.7, 121.3) | 156.22 (174.7, 137.3) | –1.23 (–2.74, 0.31) |
| Africa | 64.39 (45.03, 86.13) | 67.15 (48.42, 88.17) | –0.34 (–2.6, 1.98) | 206.18 (177.4, 234.2) | 221.26 (196.2, 247.2) | –0.64 (–1.9, 0.63) |
| Central and South Asia | 68.91 (45.31, 92.51) | 72.3 (45.6, 98.99) | –0.46 (–2.64, 1.77) | 167.69 (129.1, 206.3) | 183.48 (140.3, 226.6) | –0.84 (–2.23, 0.57) |
| East Asia | 53.44 (46.16, 60.71) | 56.11 (48.82, 63.4) | –0.68 (–3.15, 1.82) | 160.36 (130.6, 190.1) | 176.4 (146.7, 206.1) | –1.1 (–2.51, 0.33) |
| Europe | 75.53 (57.71, 93.34) | 96.66 (74.4, 119.0) | **–2.53 (–4.51, –0.51)** | 138.88 (111.6, 166.2) | 173.48 (139.1, 207.9) | **–2.26 (–3.73, –0.76)** |
| Middle East | 94.43 (75.03, 113.8) | 107.43 (88.6, 126.3) | –1.3 (–3.14, 0.57) | 198.61 (162.8, 234.4) | 222.34 (187.2, 257.5) | –1.16 (–2.43, 0.13) |
| Oceania | 48.92 (33.56, 64.27) | 53.34 (39.4, 67.27) | –0.92 (–3.51, 1.74) | 94.63 (62.2, 127.06) | 102.14 (70.5, 133.76) | –0.81 (–2.68, 1.09) |
| America | 40.44 (30.0, 52.53) | 45.86 (34.22, 59.38) | –1.24 (–4.05, 1.64) | 93.91 (80.49,105.8) | 101.2 (87.28, 114.1) | –0.7 (–2.57, 1.21) |

| Region | Ischemic heart disease | | | Intracerebral hemorrhage | | |
| --- | --- | --- | --- | --- | --- | --- |
| | 2019 | 2009 | 2009–2019 EAPC | 2019 | 2009 | 2009–2019 EAPC |
| Global | 117.95 (125.9, 107.8) | 131.77 (138.3, 122.4) | –1.17 (–2.81, 0.5) | 36.04 (38.67, 32.98) | 45.92 (48.45, 43.07) | –2.53 (–5.37, 0.39) |
| Africa | 154.04 (132.6, 175.4) | 162.34 (146.4, 180.7) | –0.45 (–1.91, 1.04) | 49.82 (43.07, 57.02) | 58.06 (51.19, 65.11) | –1.45 (–3.93, 1.09) |
| Central and South Asia | 167.39 (115.8, 219.0) | 176.75 (118.3, 235.2) | –0.51 (–1.91, 0.92) | 41.45 (32.34, 50.55) | 51.57 (37.19, 65.94) | –2.13 (–4.8, 0.62) |

*Table 1 continued on next page*

*Table 1 continued*

| | Ischemic heart disease | | | Intracerebral hemorrhage | | |
|---|---|---|---|---|---|---|
| East Asia | 110.93 (94.7, 127.17) | 117.17 (100.7, 133.7) | –0.76 (–2.47, 0.97) | 61.2 (49.06, 73.33) | 85.51 (70.7, 100.37) | **–3.54 (–5.65, –1.38)** |
| Europe | 138.19 (105.5, 170.8) | 174.31 (134.8, 213.8) | **–2.37 (–3.85, –0.88)** | 15.7 (12.25, 19.15) | 20.31 (15.85, 24.77) | –2.46 (–6.77, 2.04) |
| Middle East | 197.4 (154.9,239.9) | 221.61 (178.7, 264.5) | –1.14 (–2.41, 0.16) | 24.94 (15.45, 34.42) | 31.66 (19.43, 43.89) | –2.445 (–5.8, 1.03) |
| Oceania | 100.75 (65.96, 135.6) | 109.14 (77.5, 140.7) | –0.86 (–2.67, 0.99) | 34.32 (12.57, 56.07) | 34.25 (12.46, 56.04) | –0.056 (–3.2, 3.19) |
| America | 87.51 (79.07, 93.31) | 97.5 (89.02, 102.0) | –1.06 (–2.98, 0.91) | 14.06 (12.97, 15.03) | 15.95 (15.05, 16.68) | –1.21 (–5.93, 3.74) |

| | Ischemic stroke | | | Subarachnoid hemorrhage | | |
|---|---|---|---|---|---|---|
| Region | 2019 | 2009 | 2009–2019 EAPC | 2019 | 2009 | 2009–2019 EAPC |
| Global | 43.5 (46.8, 39.08) | 49.24 (52.21, 44.75) | –1.3 (–3.99, 1.47) | 4.66 (5.17, 4.13) | 5.34 (6.2, 4.57) | –1.29 (–9.21, 7.33) |
| Africa | 52.05 (45.89, 58.59) | 54.74 (48.9, 60.76) | –0.43 (–2.94, 2.14) | 2.73 (1.62, 4.97) | 3.27 (1.9, 5.95) | –1.71 (–11.75, 9.48) |
| Central and South Asia | 39.2 (28.7, 49.75) | 44.32 (31.11, 57.53) | –0.99 (–3.88, 1.95) | 5.77 (4.42, 7.12) | 7.15 (5.05, 9.26) | –2.16 (–9.13, 5.34) |
| East Asia | 61.33 (50.37, 72.28) | 64.96 (54.8, 75.11) | –0.734 (–3.04, 1.61) | 5.14 (4.66, 5.62) | 6.05 (5.45, 6.65) | –1.66 (–9.11, 6.4) |
| Europe | 50.36 (38.22, 62.51) | 65.67 (49.49, 81.85) | **–2.72 (–5.14,–0.25)** | 4.16 (3.55, 4.78) | 4.84 (4.08, 5.6) | –1.3 (–9.74, 7.93) |
| Middle East | 61.43 (49.71, 73.15) | 68.72 (57.62, 79.81) | –1.17 (–3.44, 1.15) | 2.91 (2.0, 3.83) | 3.74 (2.82, 4.67) | –2.39 (–11.73, 8.14) |
| Oceania | 22.56 (17.61, 27.52) | 24.92 (20.87, 28.98) | –1.04 (–4.81, 2.89) | 5.27 (3.32, 7.21) | 5.61 (3.57, 7.64) | –0.64 (–8.31, 7.68) |
| America | 21.69 (18.8, 23.4) | 23.96 (21.31, 25.49) | –0.99 (–4.83, 2.99) | 4.57 (4.16, 4.9) | 4.81 (4.47, 5) | –0.45 (–8.72, 8.55) |

| | Stroke | | | Type 2 diabetes | | |
|---|---|---|---|---|---|---|
| Region | 2019 | 2009 | 2009–2019 EAPC | 2019 | 2009 | 2009–2019 EAPC |
| Global | 84.19 (90.15, 76.76) | 100.5 (105.4, 93.24) | –1.85 (–3.76, 0.1) | 18.49 (19.66, 17.18) | 18.23 (18.97, 17.12) | 0.21 (–4.07, 4.69) |
| Africa | 104.6 (93.07, 116.9) | 116.07 (104.7, 127.9) | –0.97 (–2.71, 0.81) | 36.45 (33.0, 40.01) | 37.5 (34.54, 40.3) | –0.24 (–3.24, 2.85) |
| Central and South Asia | 86.42 (67.06, 105.8) | 103.04 (75.1, 130.94) | –1.63 (–3.53, 0.3) | 29.61 (24.29, 34.93) | 28.54 (23.45, 33.63) | 0.62 (–2.83, 4.17) |
| East Asia | 127.66 (104.9, 150.5) | 156.52 (132.0, 181.0) | **–2.2 (–3.73, –0.64)** | 17.45 (9.53, 25.37) | 16.52 (9.67, 23.37) | 0.59 (–3.85, 5.24) |
| Europe | 70.22 (54.79, 85.65) | 90.82 (70.3, 111.34) | **–2.6 (–4.65, –0.5)** | 8.26 (7.03, 9.48) | 8.46 (6.95, 9.96) | 0.09 (–6.19, 6.78) |
| Middle East | 89.28 (70.7, 107.87) | 104.12 (84.6, 123.63) | –1.59 (–3.44, 0.3) | 27.43 (19.25, 35.6) | 28.38 (18.53, 38.22) | –0.38 (–3.86, 3.22) |
| Oceania | 62.15 (34.12, 90.18) | 64.78 (37.63, 91.92) | –0.47 (–2.79, 1.9) | 41.29 (15.02, 67.55) | 40.0 (14.19, 65.81) | 0.27 (–2.65, 3.27) |
| America | 40.33 (36.24, 43.1) | 44.72 (40.98, 46.98) | –1.01 (–3.83, 1.9) | 22.55 (20.65, 24.08) | 22.89 (21.35, 23.69) | –0.08 (–3.96, 3.97) |

*Table 1 continued on next page*

*Table 1 continued*

| Region | Type 1 diabetes | | | | | |
|---|---|---|---|---|---|---|
| | 2019 | 2009 | 2009–2019 EAPC | - | - | - |
| Global | 0.98 (1.17, 0.85) | 1.03 (1.22, 0.88) | –0.45 (–17.5, 20.07) | - | - | - |
| Africa | 1.59 (1.24, 1.92) | 1.69 (1.29, 2.14) | –0.57 (–14.01, 14.9) | - | - | - |
| Central and South Asia | 1.34 (1.09, 1.6) | 1.4 (1.09, 1.71) | –0.33 (–14.93, 16.8) | - | - | - |
| East Asia | 0.91 (0.33, 1.49) | 0.98 (0.39, 1.56) | –1.17 (–18.6, 20.01) | - | - | - |
| Europe | 0.51 (0.42, 0.61) | 0.59 (0.49, 0.69) | –1.57 (–24.09, 27.6) | - | - | - |
| Middle East | 1.05 (0.8, 1.3) | 1.27 (0.97, 1.57) | –2.26 (–17.89, 16.3) | - | - | - |
| Oceania | 1.06 (0.52, 1.6) | 1.06 (0.53, 1.59) | –0.26 (–16.9, 19.74) | - | - | - |
| America | 1.04 (0.8, 1.29) | 1.1 (0.87, 1.37) | –0.27 (–16.89, 19.7) | - | - | - |

super-populations from 1000 Genome project namely, African, East Asian, South Asian, European, and American. For high BMI, the individuals clustered into three groups corresponding to African, East Asian, and European. Though broad clustering of ancestral populations among the diseases looks similar, the proportions of ancestral populations in certain diseases vary greatly. Among stroke, IHD, and T2D risk variants, the populations structured in a similar way, while for T1D, CKD, and LDL the patterns were slightly different in European and South Asian ancestry. Whereas for high SBP the major fluctuations were observed in South Asian and East Asian populations. This stratification was further explored using population-based clustering, where a similar pattern was observed in the principal component analysis (PCA) plots for stroke and its comorbid conditions (*Figure 5*). African population seems to be a distinct outlier for most diseases, and the East Asian comes a distant second in the cluster pattern.

Similar patterns were observed in stratification analysis after grouping the GWAS loci of stroke and its comorbidities into three groups as before, namely strokes, metabolic disorders, and high SBP. The African, East Asian, and South Asian populations had distinct structure in all three groups (*Figure 4— figure supplement 1*). These observations do indicate that the underlying genetic factors of stroke and its comorbid factors can be the real indicators of ethnogeographic patterns of risk for stroke and its comorbid conditions. However, we were further keen to understand the extent of shared and unique individual risk variants across stroke and its comorbid condition and how these unique or shared variants can help in distinguishing their relevance across ethnicities.

## Shared and unique risk variants of stroke among the different ethnogenetic regions

The unique and shared individual variants across stroke and its comorbid conditions were identified from the GWAS data irrespective of ethnicity. We find the majority of the risk variants were unique to a disease condition, however, several risk variants were also seen to be shared with stroke and other comorbid conditions as seen in *Figure 6*. We were further keen to have a deeper insight into the distribution of risk variants in stroke across ethnicities. Stroke has only 55% of the risk variants common to all the five populations as seen in *Figure 7* and *Supplementary file 1*. Two groups of populations share the most number of variants, namely, the Africa–America–Europe–South Asia (6% of variants shared) group and the East Asia–America–Europe–South Asia (4%) group. Africa has the highest number of unique variants for stroke (6%), followed by Europe (3%).

**Table 2.** Estimated annual percentage change (EAPC) from 2009 to 2019 of age-standardized prevalence rates (ASPRs) of stroke and its comorbid conditions.

ASPR per 100,000 for stroke and its comorbid conditions for 2009 and 2019, as well as EAPC from 2009 to 2019 is shown. 95% uncertainty interval is shown in parenthesis, statistically significant intervals are highlighted in bold.

| | Chronic kidney disease | | | High body mass index (BMI) | | |
|---|---|---|---|---|---|---|
| Region | 2019 | 2009 | 2009–2019 EAPC | 2016 | 2009 | 2009–2016 EAPC |
| Global | 8596.21 (8015.74, 9125.28) | 8340.38 (7792.87, 8846.68) | 0.34 (0.13, 0.54) | 13627.48 (14742.38, 12594.1) | 11360.96 (11989.37, 10745.18) | 2.63 (2.35, 2.91) |
| Africa | 7548.86 (7063.27, 8012.72) | 7037.32 (6575.34, 7477.98) | 0.71 (0.49, 0.93) | 12418.08 (9882.53, 14953.64) | 10068.8 (7764.52, 12373.08) | 3.04 (2.75, 3.34) |
| Central and South Asia | 8900.17 (8429.97, 9370.37) | 8656.77 (8113.18, 9200.36) | 0.33 (0.13, 0.54) | 5191.21 (3289.67, 7092.74) | 3639.99 (2090.94, 5189.04) | 5.2 (4.72, 5.69) |
| East Asia | 8868.46 (8227.29, 9509.63) | 8655.32 (8120.05, 9190.6) | 0.27 (0.07, 0.47) | 6471.83 (5527.54, 7416.12) | 4416.3 (3734.86, 5097.74) | 5.6 (5.17, 6.04) |
| Europe | 7277.72 (6612.97, 7942.47) | 7096.87 (6502.14, 7691.6) | 0.25 (0.03, 0.47) | 23696.85 (23004.53, 24389.17) | 20946.03 (20353.15, 21538.91) | 1.78 (1.57, 1.98) |
| Middle East | 10589.31 (9997.33, 11181.29) | 9904.13 (9402.68, 10405.57) | 0.69 (0.51, 0.88) | 29703.7 (26783.51, 32623.9) | 25320.15 (22564.8, 28075.51) | 2.36 (2.17, 2.55) |
| Oceania | 7282.16 (6638.89, 7925.43) | 7025.88 (6430.44, 7621.32) | 0.39 (0.17, 0.62) | 28663.27 (26270.75, 31055.79) | 24822.02 (22441.51, 27202.53) | 2.07 (1.89, 2.26) |
| America | 8795.43 (8257.8, 9295.57) | 8394.13 (7864.71, 8889.84) | 0.52 (0.31, 0.72) | 29195.14 (27054.8, 31335.48) | 25317.62 (23333.24, 27302.01) | 2.05 (1.87, 2.24) |

| | High systolic blood pressure (SBP) | | | Ischemic heart disease | | |
|---|---|---|---|---|---|---|
| Region | 2015 | 2009 | 2009–2015 EAPC | 2019 | 2009 | 2009–2019 EAPC |
| Global | 22078.6 (24798.17, 19573.65) | 23078.95 (24774.74, 21431.11) | −0.74 (−0.98, −0.49) | 2421.02 (2180.5, 2692.65) | 2422.21 (2185.13, 2688.5) | −0.01 (−0.39, 0.37) |
| Africa | 27269.91 (26515.46, 28024.36) | 27910.66 (27337.87, 28483.45) | −0.39 (−0.61, −0.16) | 2819.42 (2580.09, 3098.08) | 2814.78 (2584.93, 3072.69) | 0.02 (−0.33, 0.37) |
| Central and South Asia | 26256.46 (25139.35, 27373.58) | 26062.04 (25049.67, 27074.4) | 0.12 (−0.11, 0.35) | 3577.7 (3326.28, 3829.11) | 3510.51 (3256.78, 3764.25) | 0.18 (−0.13, 0.49) |
| East Asia | 19997.47 (18656.73, 21338.2) | 20935.67 (19829.41, 22041.94) | −0.76 (−1.02, −0.5) | 2052.09 (1817.64, 2286.54) | 2056.98 (1821.18, 2292.77) | −0.01 (−0.43, 0.4) |
| Europe | 23278.69 (21812.71, 24744.67) | 25915.06 (24642.41, 27187.72) | −1.77 (−2, −1.54) | 2476.75 (2203.31, 2750.18) | 2548.31 (2267.8, 2828.82) | −0.3 (−0.67, 0.08) |
| Middle East | 22293.46 (20587.65, 23999.28) | 24327.72 (23042.57, 25612.87) | −1.48 (−1.72, −1.24) | 4843.02 (4243.46, 5442.58) | 4949.35 (4386.28, 5512.41) | −0.27 (−0.54, −0.003) |
| Oceania | 18081.14 (15851.15, 20311.13) | 19610.27 (18278.28, 20942.26) | −1.34 (−1.61, −1.08) | 2734.98 (2611.03, 2858.93) | 2843.93 (2697.13, 2990.73) | −0.43 (−0.79, −0.08) |
| America | 18041.91 (16571.13, 19512.69) | 19505.67 (17874.04, 21137.3) | −1.29 (−1.56, −1.02) | 1695.56 (1530.76, 1871.92) | 1770.96 (1602.8, 1955.71) | −0.46 (−0.91, −0.01) |

| | Intracerebral hemorrhage | | | Ischemic stroke | | |
|---|---|---|---|---|---|---|
| Region | 2019 | 2009 | 2009–2019 EAPC | 2019 | 2009 | 2009–2019 EAPC |
| Global | 248.77 (217.09, 281.43) | 257.65 (226.36, 289.73) | −0.4 (−1.57, 0.79) | 950.97 (849.82, 1064.06) | 916.46 (827.21, 1011.45) | 0.43 (−0.18, 1.05) |
| Africa | 314.25 (281.2, 349.75) | 340.25 (305.47, 377.42) | −0.88 (−1.9, 0.15) | 1089.33 (984.73, 1198.86) | 1071.4 (971.23, 1174.89) | 0.18 (−0.39, 0.75) |
| Central and South Asia | 267.03 (223.81, 310.25) | 268.32 (214.19, 322.45) | −0.05 (−1.19, 1.1) | 570.43 (480.32, 660.53) | 533.37 (437.92, 628.82) | 0.76 (−0.04, 1.57) |

*Table 2 continued on next page*

*Table 2 continued*

| | Intracerebral hemorrhage | | | Ischemic stroke | | |
|---|---|---|---|---|---|---|
| | 2019 | 2009 | 2009–2019 EAPC | 2019 | 2009 | 2009–2019 EAPC |
| East Asia | 302.65 (221.3, 384.01) | 319.25 (236.89, 401.6) | −0.58 (−1.64, 0.48) | 1183.05 (1086.71, 1279.4) | 1074.86 (1001.51, 1148.22) | 1.09 (0.53, 1.65) |
| Europe | 139.23 (120.98, 157.49) | 158.58 (135.99, 181.17) | −1.44 (−2.94, 0.09) | 753.15 (676.12, 830.17) | 813.8 (730.34, 897.26) | −0.85 (−1.51, −0.19) |
| Middle East | 255.65 (199.55, 311.75) | 268.51 (211.97, 325.06) | −0.7 (−1.85, 0.46) | 1218.88 (1060.37, 1377.4) | 1220.26 (1066.3, 1374.22) | −0.07 (−0.61, 0.47) |
| Oceania | 264.45 (113.71, 415.19) | 258.3 (103.93, 412.68) | 0.17 (−0.98, 1.34) | 650.92 (557.27, 744.58) | 655.29 (569.12, 741.47) | −0.09 (−0.82, 0.64) |
| America | 200.52 (176.38, 226.00) | 212.00 (188.08, 237.72) | −0.58 (−1.87, 0.73) | 932.69 (829.99, 1041.00) | 977.11 (888.38, 1068.44) | −0.4 (−1.00, 0.21) |

| | Subarachnoid hemorrhage | | | Stroke | | |
|---|---|---|---|---|---|---|
| Region | 2019 | 2009 | 2009–2019 EAPC | 2019 | 2009 | 2009–2019 EAPC |
| Global | 101.57 (87.13, 118.54) | 103.6 (89.2, 120.53) | −0.22 (−2.05, 1.64) | 1240.26 (1139.71, 1352.99) | 1217.36 (1126.81, 1313.52) | 0.22 (−0.31, 0.76) |
| Africa | 52.47 (44.76,61.46) | 53.58 (45.9, 62.63) | −0.2 (−2.74, 2.39) | 1393.27 (1288.31, 1503.93) | 1402.18 (1303.49, 1509.44) | −0.08 (−0.57, 0.43) |
| Central and South Asia | 65.01 (56.51,73.52) | 64.76 (54.84,74.67) | 0.05 (−2.25, 2.41) | 858.5 (737.58, 979.42) | 822.67 (682.76, 962.57) | 0.48 (−0.16, 1.14) |
| East Asia | 107.87 (72.53, 143.2) | 107.03 (72.44, 141.62) | 0.12 (−1.67, 1.95) | 1513.1 (1390.7, 1635.49) | 1425.39 (1298.41, 1552.38) | 0.69 (0.2, 1.18) |
| Europe | 118.34 (108.49, 128.2) | 124.24 (112.18, 136.31) | −0.52 (−2.19, 1.18) | 960.86 (861.79, 1059.92) | 1042.89 (932.11, 1153.68) | −0.9 (−1.49, −0.31) |
| Middle East | 66.69 (58.12,75.25) | 67.19 (57.51,76.87) | −0.03 (−2.27, 2.26) | 1471.6 (1292.03, 1651.17) | 1485.6 (1313.34, 1657.86) | −0.18 (−0.66, 0.31) |
| Oceania | 105.79 (85.55, 126.03) | 104.66 (83.96, 125.35) | 0.13 (−1.68, 1.97) | 970.31 (719.9, 1220.73) | 967.25 (721.78, 1212.72) | −0.0006 (−0.6, 0.6) |
| America | 138.9 (119.01, 162.64) | 127.3 (110.52, 146.72) | 0.75 (−0.87, 2.4) | 1215.6 (1113.69, 1328.23) | 1259.25 (1171.81, 1351.47) | −0.32 (−0.85, 0.22) |

| | Type 1 diabetes | | | Type 2 diabetes | | |
|---|---|---|---|---|---|---|
| Region | 2019 | 2009 | 2009–2019 EAPC | 2019 | 2009 | 2009–2019 EAPC |
| Global | 272.54 (216.98, 336.95) | 250.88 (203.63, 307.25) | 0.71 (−0.45, 1.89) | 5282.85 (4853.59, 5752.09) | 4646.97 (4341.45, 4960.45) | 1.24 (0.97, 1.51) |
| Africa | 255.51 (201.1, 318.05) | 227.8 (179.02, 286.18) | 1.16 (−0.06, 2.39) | 4516.3 (4096.49, 4976.22) | 3939.47 (3629.16, 4276.48) | 1.35 (1.06, 1.64) |
| Central and South Asia | 250.87 (235.66, 266.09) | 222.68 (211.11, 234.25) | 1.12 (−0.1, 2.36) | 6401.5 (5931.41, 6871.58) | 5054.2 (4714.43, 5393.97) | 2.18 (1.93, 2.44) |
| East Asia | 132.14 (92.27, 172) | 117.02 (75.77, 158.27) | 0.97 (−0.72, 2.7) | 4477.09 (4223.96, 4730.21) | 4237.06 (3842.25, 4631.87) | 0.68 (0.4, 0.97) |
| Europe | 499.9 (451.46, 548.34) | 458.58 (394.14, 523.02) | 0.69 (−0.17, 1.56) | 4691.56 (4113.82, 5269.3) | 4044.18 (3599.81, 4488.54) | 1.45 (1.16, 1.74) |
| Middle East | 394.02 (354.45, 433.59) | 297 (246.27, 347.72) | 2.84 (1.79, 3.89) | 6796.8 (5773.01, 7820.59) | 5937.92 (5063.13, 6812.7) | 1.3 (1.06, 1.53) |
| Oceania | 459.51 (335.35, 583.67) | 441.02 (327.93, 554.11) | 0.4 (−0.48, 1.29) | 5606.6 (3790.83, 7422.38) | 4732.14 (3129.63, 6334.64) | 1.87 (1.61, 2.14) |
| America | 417.47 (337.43, 511.23) | 408.22 (340.36, 487.84) | 0.14 (−0.79, 1.07) | 6563.01 (6071.37, 7036.07) | 5862.85 (5515.86, 6198.91) | 1.09 (0.85, 1.33) |

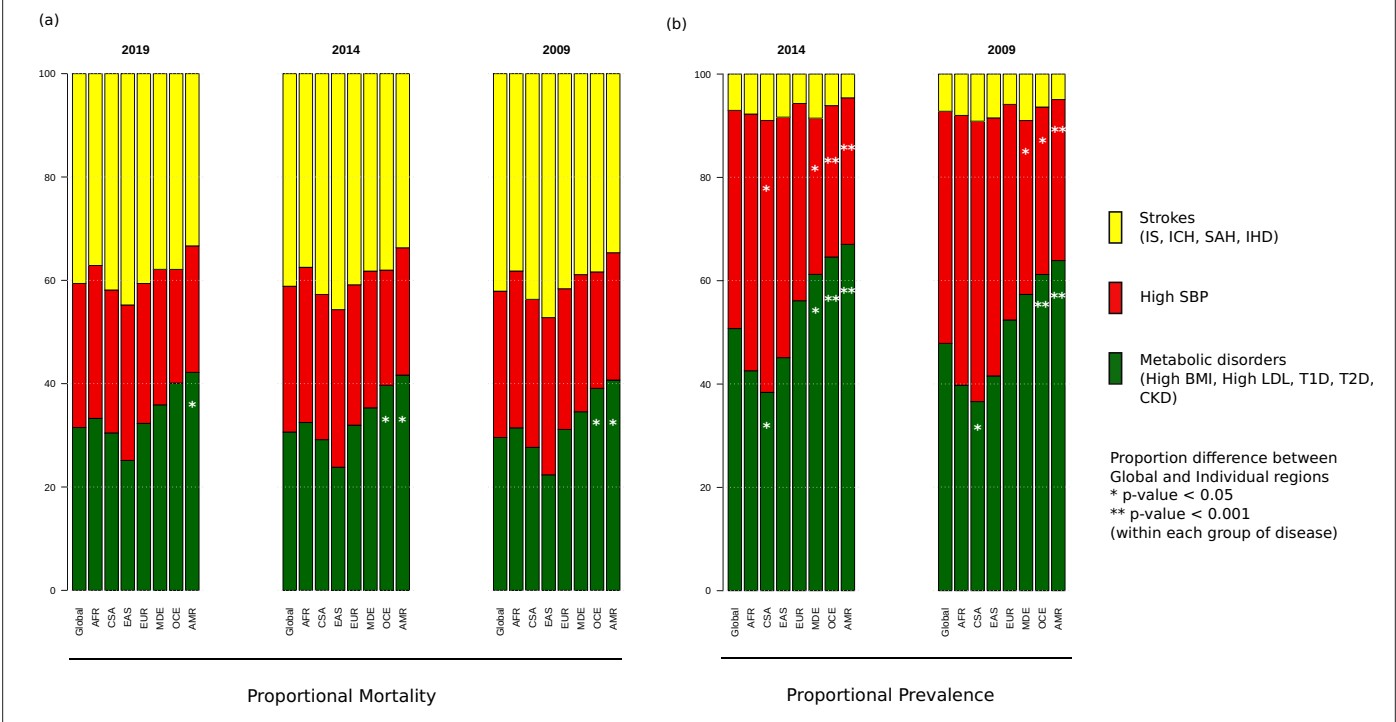

**Figure 2.** Proportional (**a**) mortality rates and (**b**) prevalence rates for strokes, high systolic blood pressure (SBP), and metabolic disorders. The figure shows the proportional mortality and prevalence for all strokes in yellow (ischemic stroke, intracerebral hemorrhage, subarachnoid hemorrhage, and ischemic heart disease), high SBP in red, and metabolic disorders in green (high body mass index [BMI], high low-density lipoprotein [LDL], diabetes mellitus type 1 and 2, chronic kidney disorder). Proportions significantly different from global proportions are marked with asteriks. CSA – Central and South Asia, AFR – Africa, EAS – East Asia, EUR – Europe, MDE – Middle East, OCE – Oceania, AMR – America.

South Asia has two unique variants for stroke, rs528002287 and rs148010464, which maps to genes PCSK6 and the intergenic region of PLA2G4A/LINC01036, respectively. The variants rs528002287 and rs148010464 are low-frequency variants in South Asia with a minor allele frequency (MAF) of 0.053 and 0.054, respectively. We were further keen to understand how these unique variants in South Asia are tagged to the nearby variants of different frequencies in the different ethnicities. Can the linkage disequilibrium (LD) patterns of rare allele and common allele help in distinguishing the ethnogeographic distinction in phenotype variation. While comparing, the LD pattern of low-frequency variants and common variants across ethnicities, we observe contrasting patterns. The LD plots between the unique variants and low-frequency variants, that are tagged to it, clearly demonstrated a unique LD pattern in South Asia, compared to other populations (*Figure 8*). Contrastingly, the LD plots of common variants tagged to the risk variant of stroke unique to South Asia showed similar LD patterns among all populations (*Figure 8—figure supplement 1*). These differences might also reflect unique or distinct phenotypic differences among ethnicities for risk in stroke.

## Discussion

To the best of our knowledge, this is the first study that explored the burden of stroke and its comorbid conditions across regions, stratifying and distinguishing their unique features based on their genetic background extrapolated from the GWAS risk loci. The dynamics of different rates of stroke, its subtypes, and comorbid factors do reflect ethnogeographic differences. Globally, the prevalence and incidence rates of stroke have increased, while mortality rates decreased with minor shifts in ranking in the last decade. Interestingly, the incidence and prevalence rank of stroke rates remain the same globally, but for mortality it ranks third, preceded only by IHD and high SBP. While the global stroke prevalence is nearly 15 times its mortality rate, prevalence of comorbid conditions such as high SBP, high BMI, CKD, and T2D are alarmingly 150- to 500-fold higher than their mortality rates. These comorbid conditions can drastically affect the outcome of stroke. Interestingly, these disparities in rates get

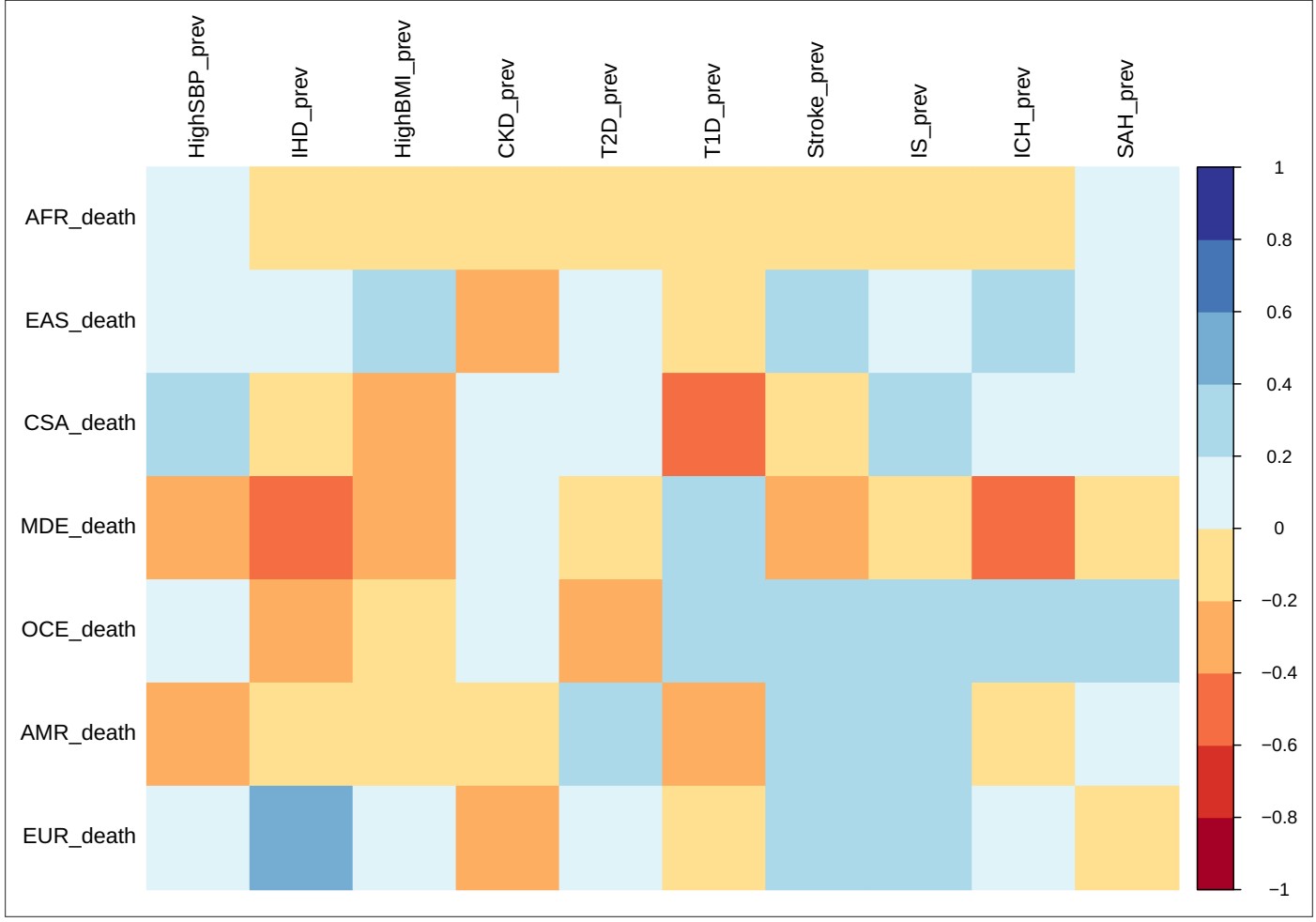

**Figure 3.** Correlation of mortality versus prevalence rates of stroke, its subtypes, and comorbid factors. The figure shows the Pearson correlation coefficients of age-standardized mortality rates versus prevalence rates across different continents for 2014. CSA – Central and South Asia, AFR – Africa, EAS – East Asia, EUR – Europe, MDE – Middle East, OCE – Oceania, AMR – America.

further widened when evaluated from an ethnogeographic perspective. The age-standardized prevalence of stroke in 2019 ranges from lowest in Central and South Asia (858.5/100,000, 95%UI 737.6–979.4) to highest in East Asia (1513.1/100,000, 95%UI 1390.7–1635.5), in contrast to mortality rates, lowest in America (40.3/100,000, 95%UI 36.2–43.1) and highest in East Asia (127.7/100,000, 95%UI 104.9–150.5).

The rates of stroke, its subtypes, and comorbid conditions do correlate to some extent but their ranking varies significantly. In terms of ranking, stroke ranks sixth in prevalence across all ethnogeographic locations, but ranks second in mortality in East Asia and third in Central and South Asia and Africa. We find that among the considered comorbid conditions, some rank above stroke in both incidence and prevalence, however in terms of mortality, stroke ranks highest with exception to high SBP. Similarly, the ranking of the comorbid conditions also varies when the global population is stratified based on ethnogeographic locations. In the last decade, there has been tremendous development in the healthcare industry globally, but this is not reflected in the mortality or prevalence data from 2009 to 2019. Ranking of comorbid conditions by the rates is very crucial to identify ethnic-specific comorbid risk that can be helpful in guiding and managing stroke risk.

The changing dynamics of stroke or its comorbid conditions can be attributed to a multitude of factors. Often the global burden of stroke has been discussed from the point of view of socio-economic parameters. Studies indicate that half of the stroke-related deaths are attributable to poor management of modifiable risk factors (*Avan et al., 2019*; *Baatiema et al., 2020*). However, we observe that different socio-economic regions are driven by different risk factors. Considering that

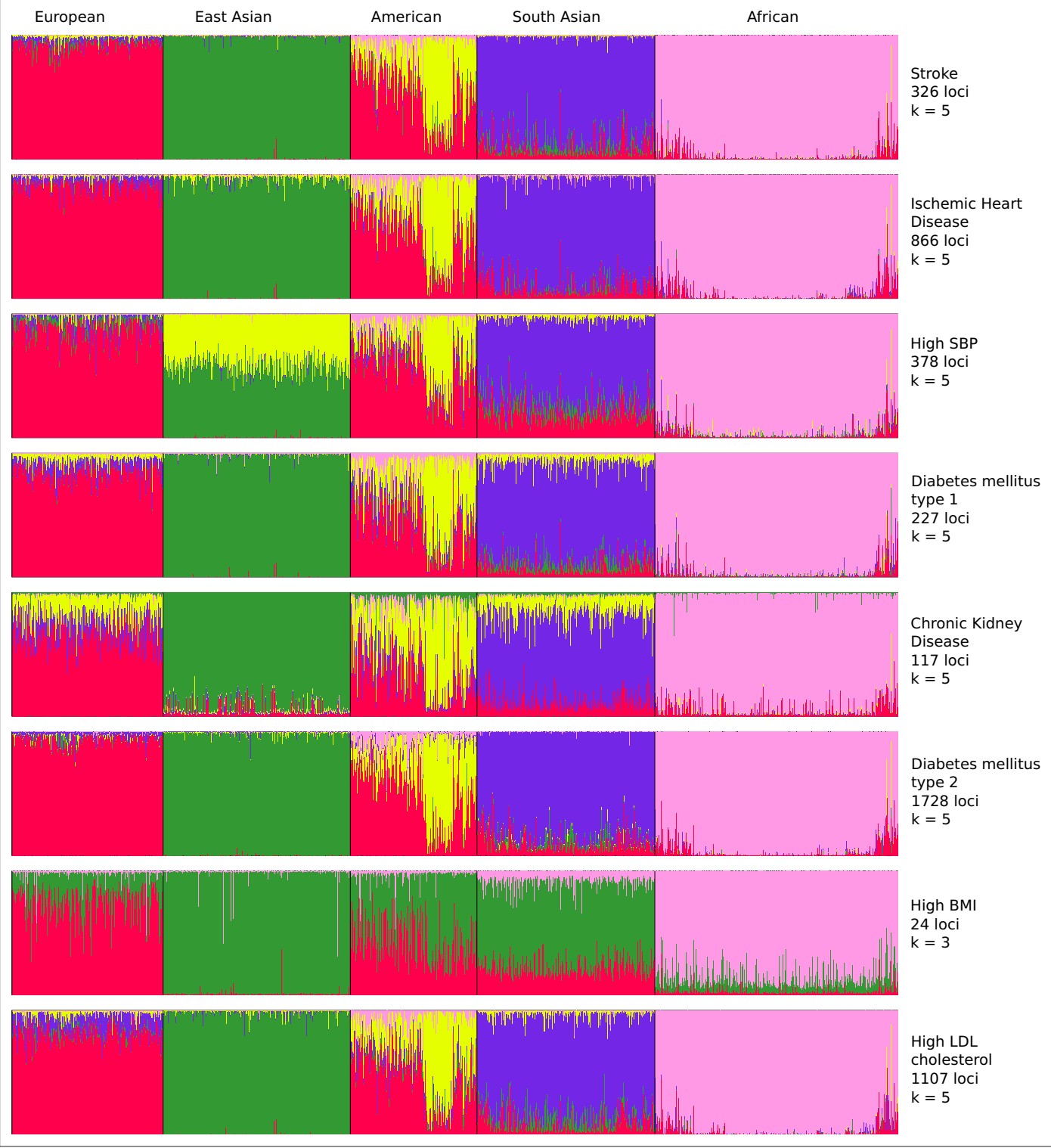

**Figure 4.** Population structure of risk variants for stroke and its comorbid factors using a model-based clustering. The proportion of ancestral populations was estimated from the genotype of risk variants of each disease in unrelated individuals from the 1000 Genomes project using a model-based clustering approach. The individuals were represented by their super-population in 1000 Genomes (African, East Asian, South Asian, European, and American). The estimated ancestral populations clustered into either five or three clusters (in case of body mass index [BMI]) represented by the different colors – red (population cluster 1), green (population cluster 2), yellow (population cluster 3), purple (population cluster 4), and pink (population cluster 5).

The online version of this article includes the following figure supplement(s) for figure 4:

*Figure 4 continued on next page*

*Figure 4 continued*

**Figure supplement 1.** Population structure of risk variants for strokes, high systolic blood pressure (SBP), and metabolic disorders using a model-based clustering.

Europe, America, Oceania, and the Middle East represent high socio-economic regions, the comorbid conditions that drive prevalence and mortality rates here seem to be more of metabolic in nature, while for South Asians, high SBP is the prominent factor. It is evident from the correlation between prevalence and mortality for stroke, its subtypes, and comorbid conditions that there is an ethnic- and comorbid-specific correlation, which possibly does not reflect a clear socio-economic distinction. The

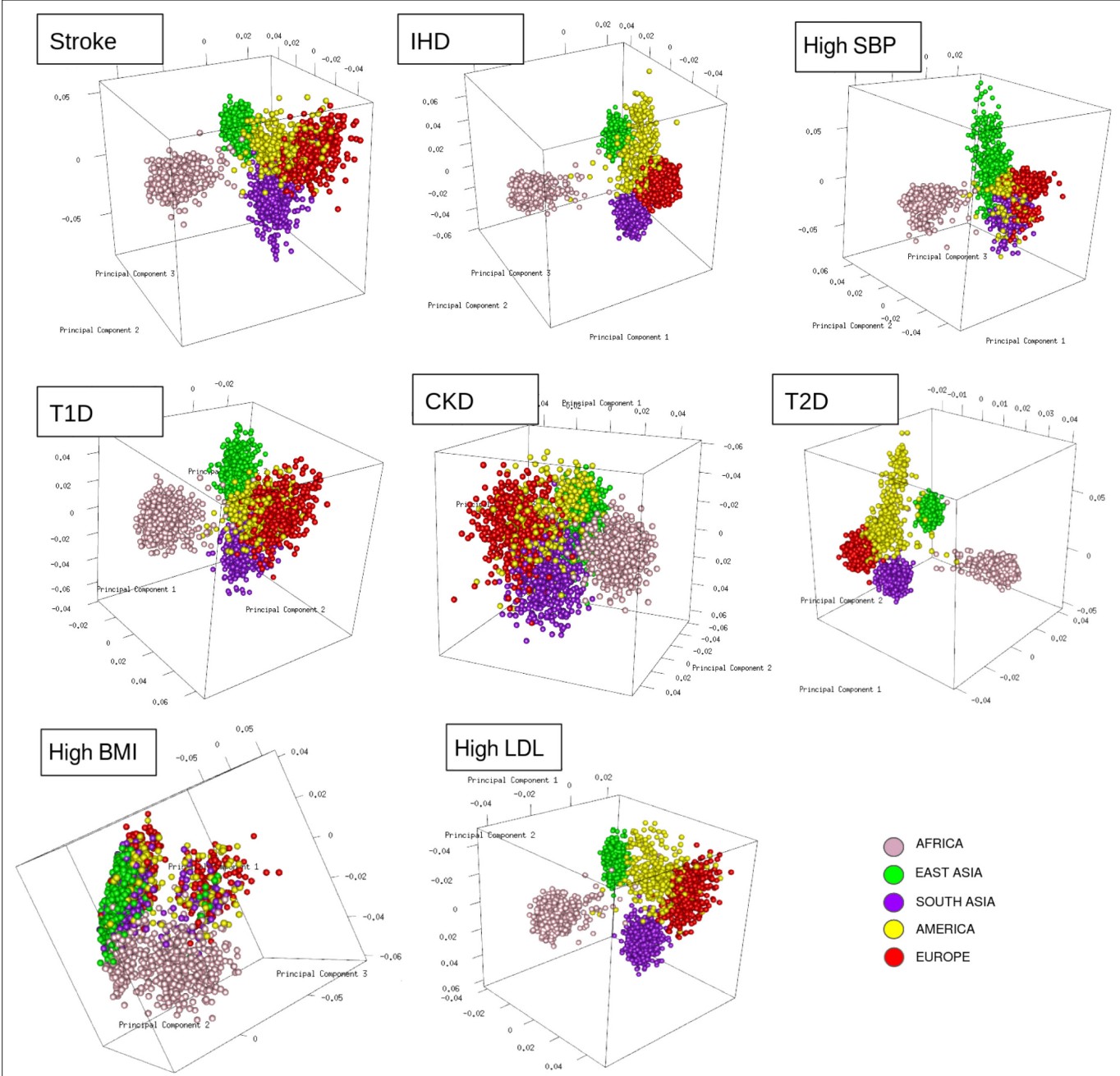

**Figure 5.** Clustering of risk variants for stroke and its comorbid factors using a PCA. The eigenvectors were estimated from the genotype of risk variants of each disease in unrelated individuals from the 1000 Genomes project using PCA clustering. The individuals were represented by their super-population in 1000 Genomes (African, East Asian, South Asian, European, and American) shown in five different colors.

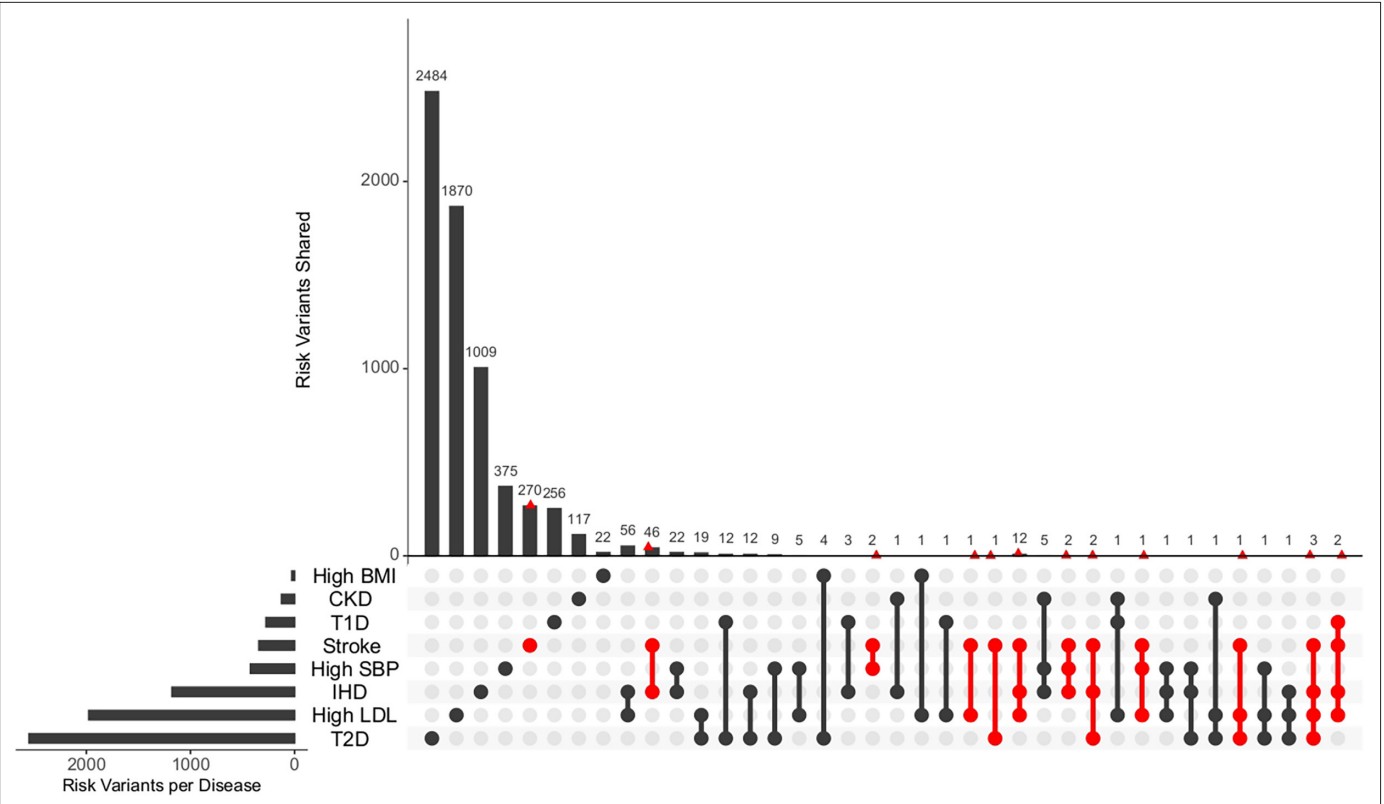

**Figure 6.** Distribution of unique and shared risk variants in stroke and its comorbid conditions. The bar plot shows the number of GWAS risk variants unique for stroke and its comorbidities, as well as number of GWAS risk variants shared among the different diseases. The intersection of diseases is indicated below. Intersections containing stroke are highlighted in red. The total number of GWAS risk variants considered is stroke (366), ischemic heart disease (IHD) (1137), high systolic blood pressure (SBP) (419), type 2 diabetes (T2D) (2227), type 1 diabetes (T1D) (269), chronic kidney disease (CKD) (125), high body mass index (BMI) (27), and high low-density lipoprotein (LDL) (1734).

comorbid conditions for stroke subtypes also differ to a large extent. Diabetes is a comorbid condition for stroke but not for SAH, and this risk is ethnic-specific (*Lindgren et al., 2013*; *Koshy et al., 2010*). Therefore, it is very pertinent to understand the stroke risk from an ethnic view point, beyond the boundaries of socio-economic criteria, as the drivers of comorbid risk and ethnicity rely on genetic and epigenetic components. Studies reported reduction in life expectancy in 31 of 37 high-income countries, deduced to be due to COVID-19 (*Islam et al., 2021*). However, it would be unfair to ignore the comorbid conditions which could also be the critical determinants for reduced life expectancy in these countries.

Stroke has a complex etiology, which is further influenced by its comorbid conditions and this impacts its phenotypic variability. A strong genetic risk drives both stroke and its comorbid conditions. Genetic risk variants for diabetes, cardiovascular disease, diabetic retinopathies and nephropathies, hypertension, inflammation, and kidney diseases have been reportedly shown to have strong ethnogenetic variation (*Shoily et al., 2021*). Implications of ethnogenetic differences were evident when GWAS genes for stroke and all studies of comorbid conditions were used to stratify the 1000 genome super-populations. We observed that the GWAS risk loci for stroke and its comorbid conditions like high BMI, high LDL, high SBP, T2D, T1D, and CKD could stratify the super-populations based on its ethnogenetic considerations. Fluctuations in genetic structure of stroke and its comorbid conditions signify the impact of ethnic variations on mortality and prevalence rates. Stroke accounts for approximately 20% of deaths in diabetics (*Banerjee et al., 2012*; *Boehme et al., 2017*). Diabetics and prediabetics, and the duration of diabetes have been reported to have increased risk of stroke, which gets aggravated in African-Americans (*Banerjee et al., 2012*; *Boehme et al., 2017*). As stroke and its comorbid conditions are heavily influenced by lifestyle, high-income countries showed evidence of metabolic disorders being the major cause of concern for both prevalence and mortality. Similar

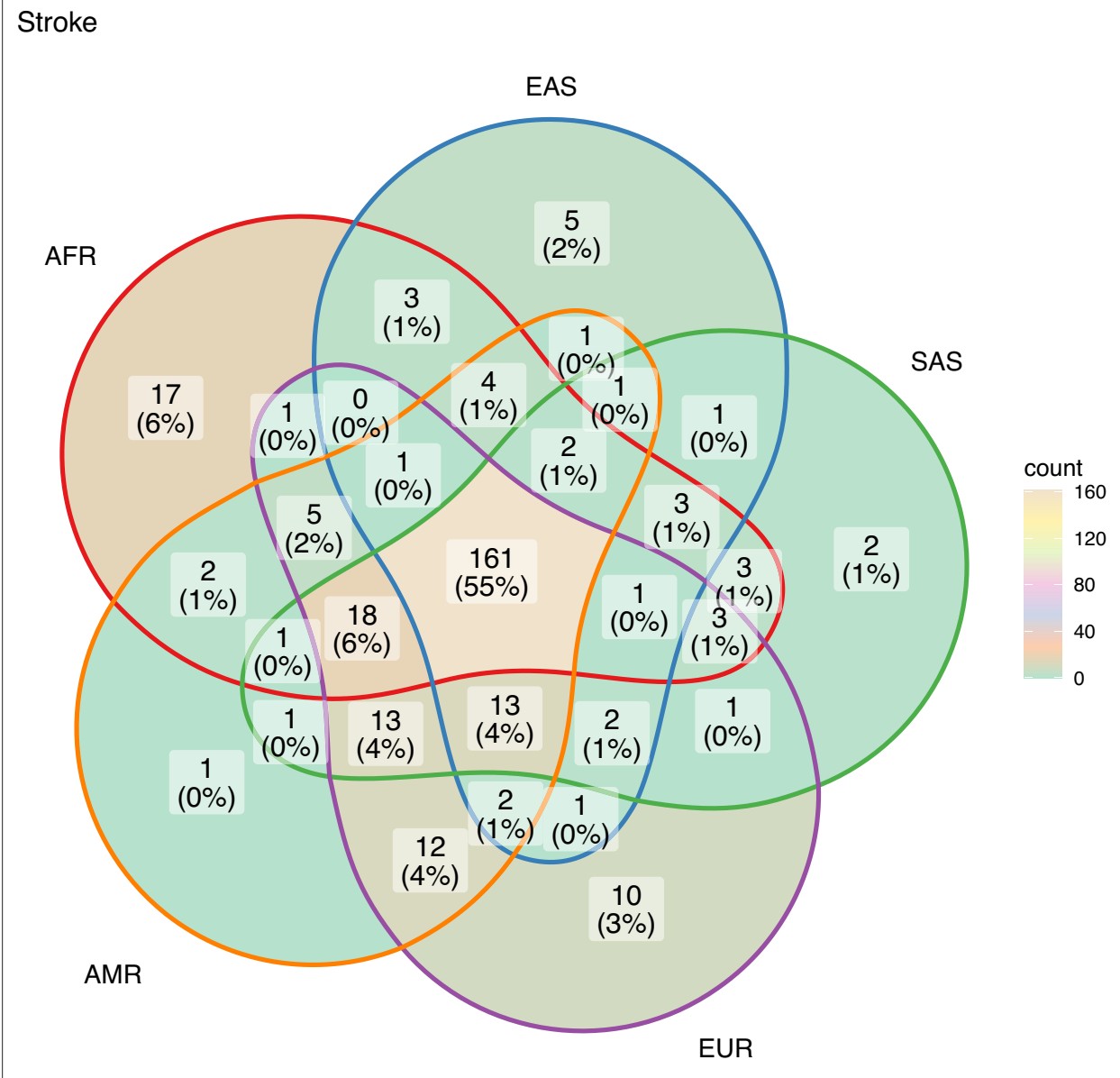

**Figure 7.** Risk variants of stroke shared among super-populations. The GWAS risk variants for stroke (total 291) shared among the five super-populations in 1000 Genomes (African, East Asian, South Asian, European, and American). A risk variant was considered to be present in a population if the alternate allele frequency in 1000 Genomes was greater than or equal to 0.05.

observation in a UK biobank cohort study on stroke suggests that genetic and lifestyle factors were independently associated with incident stroke, which emphasizes the benefit of entire populations adhering to a healthy lifestyle, independent of genetic risk (*Rutten-Jacobs et al., 2018*).

Metabolic risk variations could also be a reflection of their underlying genetic differences. Significant differences among ethnicities in metabolism of various macromolecules have been reported (*Vasishta et al., 2022*). Genetic variations demonstrate inter-ethnic differences in LDL levels resulting in differential impact on dyslipidemia (*Klarin et al., 2018*). A meta-analysis on European, East Asian, and African-American ethnicities revealed that common variants of *CDH13* and *ADIPOQ* regulate adiponectin levels, an important component of BMI indicator (*Dastani et al., 2012*). *ALDH2*504Lys* allele has been reported to be associated with high BMI, increased tolerance of alcohol, high SBP, and decreased high-density lipoprotein in East Asians (*Takeuchi et al., 2011*; *Xu et al., 2010*). MEGA-STROKE consortium on stroke and its comorbid factors reported five variants associated with blood pressure, two with LDL cholesterol and reported that all stroke subtypes were associated with a

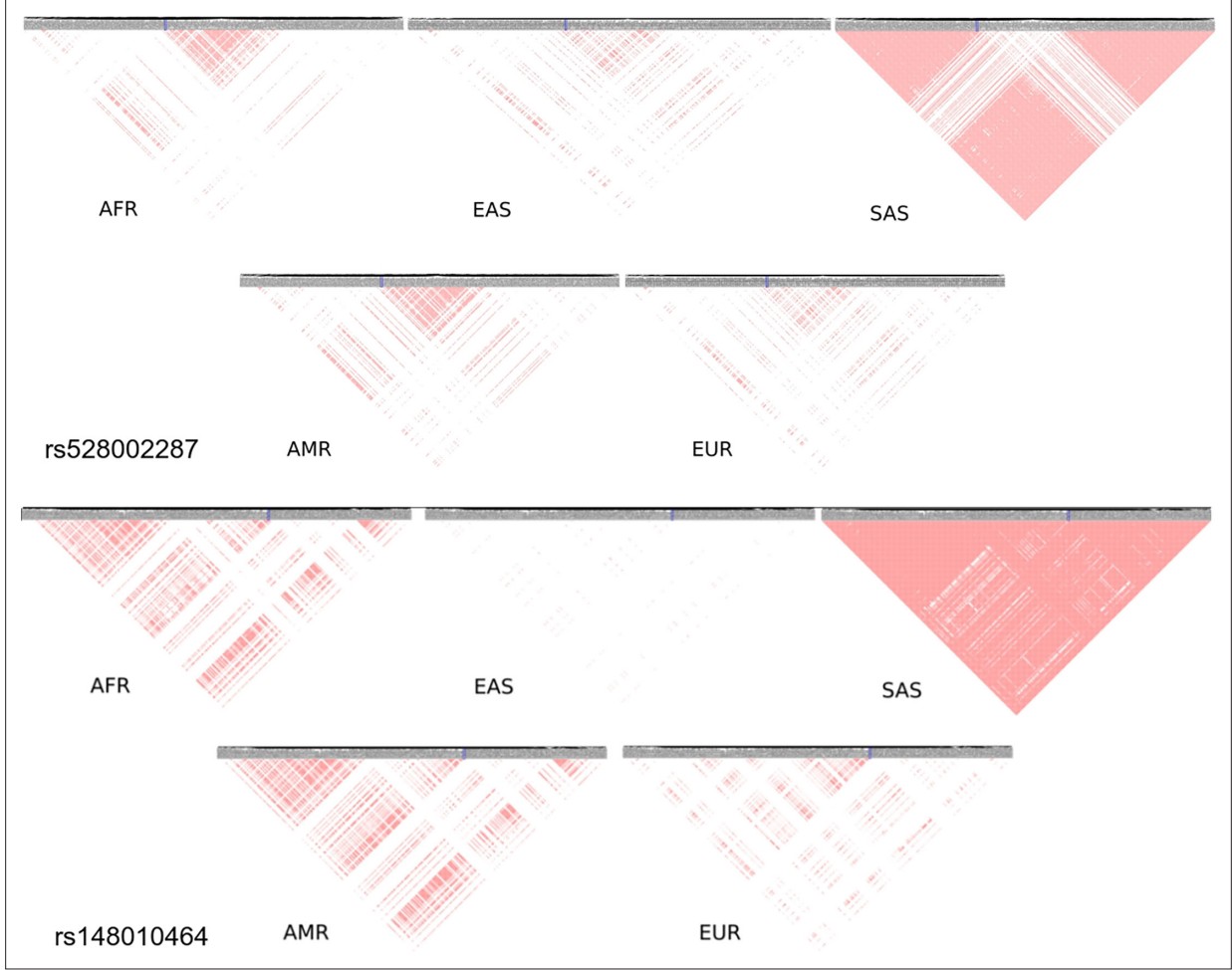

**Figure 8.** Linkage disequilibrium (LD) of low-frequency variants proxy to risk variants of stroke unique in South Asian population. Proxy variants with frequency less than 0.1 in 1 MB region flanking the stroke risk variants unique in South Asia rs528002287 and rs148010464. The risk variants are marked in blue box. LD plots of variants in different 1000 Genome super-populations (AFR – Africa, EAS – East Asia, SAS – South Asia, AMR – America, EUR – Europe) are shown.

The online version of this article includes the following figure supplement(s) for figure 8:

**Figure supplement 1.** Linkage disequilibrium of common variants proxy to risk variants of stroke unique in South Asian population.

Genetic Risk Score for high SBP (*Malik et al., 2018*). A study on IHD, T2D in European populations with different healthcare systems, and local population substructures reported Polygenic risk score with similar accuracy across Europeans, to a lesser extent to South and East Asian populations, and very poor transferability for Africans (*Mars et al., 2022*).

Even stroke subtypes show a strong ethnic variation (*Malik et al., 2018*; *Hu et al., 2022*). While *COL4A1* and *COL4A2* were common denominators for most of the stroke subtypes, high-density lipoprotein was inversely associated with small vessel stroke (*Meschia, 2020*). The INTERSTROKE study on stroke subtypes reported that among comorbid factors, high SBP was significantly associated with ICH and diabetes, cardiac issues, and apolipoproteins with ischemic stroke (*O'Donnell et al., 2016*). Ischemic stroke associated with ApoB/ApoA1 ratio had higher (67.6%) population attributable fraction (PAF) in Southeast Asia compared to Western Europe, North America, and Australia (24.8%), while ischemic stroke associated with atrial fibrillation had lower PAF in South Asia (3.1%) compared to the rest (17.1%) (*O'Donnell et al., 2016*). Emerging studies like novel ethnic-specific genetic variants, such as SUMOylation pathway in Indians (*Kumar et al., 2021*), *SFXN4* and *TMEM108* in Africans (*Keene et al., 2020*) indicate the involvement of different pathways among different ethnicities in stroke. The A allele of c.*84G>A loci in *CETP* gene was found to be a risk factor for IHD in South Asians (*Ganesan et al., 2016*). High SBP was found to be a risk factor in all major stroke subtypes

except lobar ICH (*Georgakis et al., 2020*). Therefore, identifying differential risk in different ethnicities for stroke and its subtypes, and its impact on comorbid conditions might also indicate different treatment modalities which can minimize adverse metabolic side effects.

Identifying the pattern of genetic variation is critical in distinguishing stroke and its endophenotypic variations. The risk variants rs528002287 (locus 15q26.3) in *PCSK6* and rs148010464 (locus 1q31.1) an intergenic variant in *PLA2G4A/LINC01036* for stroke were unique to South Asia, and were found to be associated with cardioembolic stroke and small vessel stroke in South Asians (*Kumar et al., 2021*). A recent INTERSTROKE study reported the association of short sleep duration with increased risk for stroke to be highest in the South Asian ethnicity (Odds Ratio 9.13, 95% Confidence Interval 5.86–14.66) (*Mc Carthy et al., 2023*). Decreased sleep quantity and quality have been reported to increase blood pressure (*Sabanayagam and Shankar, 2010*), prevalence for which was found to be highest for South Asians in our study. These observations are interesting as *PCSK6* is known to regulate sodium homeostasis and thereby maintain diastolic blood pressure (*Li et al., 2004*; *Chen et al., 2015*). Reports also indicate 2.34-fold difference in the expression of *PCSK6* during maintenance phase of hypertension (*Marques et al., 2010*). PCSK6 has also been reported to be involved in processing of precursors of Melanin Concentrating hormone (MCH) under certain conditions (*Viale et al., 1999*). MCH is known to play a central role in promoting and stabilizing sleep (*Jego et al., 2013*; *Konadhode et al., 2013*). In insomnia patients, *PLA2G4A* was reported to be upregulated by 1.88-fold after improvement in sleep (*Livingston et al., 2015*). In sleep deprived mice, glycolytic pathway and lipid metabolism were upregulated and expression of *PLA2G4A* was downregulated (*Hinard et al., 2012*). These observations are interesting as *PLA2G4A* is known to play a role in the metabolism of phospholipids, production of lipid mediators, and the release of arachidonic acid (*Burke and Dennis, 2009*; *Shimizu et al., 2006*). Arachidonic acid is involved in signaling pathways of metabolic processes like release of insulin and glucose disposal (*Chen et al., 1996*; *Nugent et al., 2001*; *Wolford et al., 2003*). *PLA2G4A* also plays a role in the production of pro-thrombotic TXA2 and thus, inhibition of *PLA2G4A* can reduce platelet aggregation and thromboembolism (*Murakami et al., 2011*). Thus, the risk of unique genetic variants in *PCSK6* and *PLA2G4A* in South Asian ancestry may indicate a unique endophenotype for stroke, which might also indicate the influence of underlying risk variants for comorbid conditions, for example *PLA2G4A* in metabolic processes.

GWAS has yielded numerous common risk alleles that are associated with various human phenotypes (*Manolio et al., 2009*; *McCarthy and Hirschhorn, 2008*). Since the rationale for GWAS is the 'common disease, common variant' hypothesis, it has been able to identify only those common variants with a moderate effect on the associated trait and thus, these identified variants only explain a small proportion of the heritability of the trait. One of the major conclusions from the 1000 Genomes Project was that most variations in the human genome are rare and unique to specific subpopulations (*Abecasis et al., 2010*; *Abecasis et al., 2012*). From an evolutionary point of view, alleles with strong effects that are detrimental will be controlled by purifying selection keeping its frequency low. Hence, rare and low-frequency variants might be variants with large effects on traits (*Bodmer and Bonilla, 2008*). Examples of genes like *ABCA1*, *PCSK9*, and *LDLR* (*Kathiresan et al., 2009*; *Lusis and Pajukanta, 2008*), which carries both common variants with moderate effects as well as rare variants with large effect for lipid levels indicate that genes can contain both types of variants associated with a complex trait. Using this logic, we were keen to identify the influence of common and rare variants in stroke and its comorbid conditions and their pattern of LD in distinguishing ethnic-specific risk.

While we observe the majority of the GWAS variants associated with stroke are common variants, a minority of these are low-frequency variants with below 10% frequency. The alternate alleles of these variants were found to be present only in specific super-populations of 1000 Genomes, while the variants are monoallelic in the other super-populations. This difference among populations gets further enhanced when we look at the LD patterns of these low-frequency variants with other low-frequency variants which are in proxy. Distinct LD blocks with these unique rare variants present in one super-population were seen to be absent in other populations. On the other hand, LD patterns of common variants (frequency greater than 10%) in the same regions show similar LD patterns for all populations. Thus, the rare variant hypothesis could explain a significant proportion of the differences in burden of stroke seen across populations. Such genes identified could be possible candidate genes for identifying rare and low-frequency variants that could play a role in the heritability of stroke and its comorbid factors.

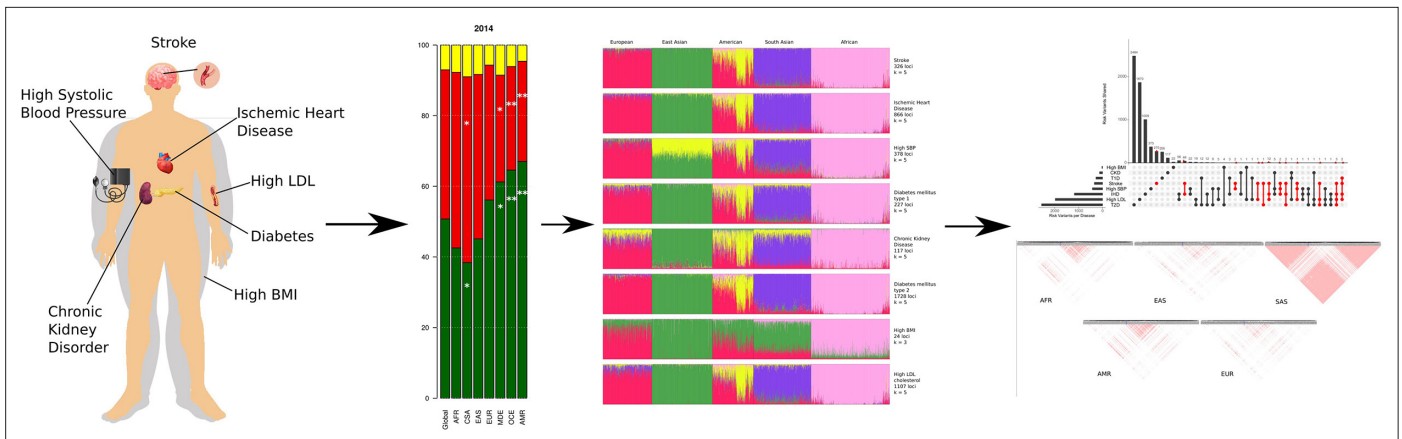

**Figure 9.** Summary of the work highlighting that the burden of stroke is driven by different risk factors in different regions. The different socio-economic regions are driven by different risk factors of stroke and low-frequency genetic variants could be playing a role in the differences in burden of stroke seen in the different regions.

## Conclusion

The dynamics of incidence, prevalence, and mortality rates of stroke and its subtypes along with its comorbid risk factors reflect strong ethnogeographic differences. Our work highlights that these ethnogeographic differences for stroke and its comorbid conditions need to be evaluated and stratified based on their ethnogenetic background. Genetic variables should be considered as primary evidence as they define the threshold for biochemical, metabolomic, or epigenetic variables. The different socio-economic regions are driven by different risk factors of stroke and low-frequency variants could be playing a role in the differences in burden of stroke seen in the different regions as shown in *Figure 9*. Identifying population-specific unique variants for stroke and its comorbid conditions might refine the drivers for endophenotypic variations for stroke risk. We would like to suggest that integrating public health genomics and articulating it with comorbid conditions for stroke should be considered crucial irrespective of the economic status, as both lower and higher socio-economic regions have different drivers of stroke risk.

## Methods

### Data sources

We obtained ASIRs, ASPRs, and ASMRs for a total of eight diseases and three disease conditions in 204 countries, for the years 2009–2019 using the GBD Results Tool (vizhub.healthdata.org/gbd-results/) (*Global Burden of Disease Collaborative Network, 2020*) and from NCD Risk Factor Collaboration (NCD-RisC, ncdrisc.org/) study (*Zhou et al., 2017*; *NCD Risk Factor Collaboration NCD-RisC, 2017*) in November 2021. GBD codes for the selected diseases were B.2.2 (Ischemic heart disease, IHD), B.2.3 (Stroke), B.2.3.1 (Ischemic stroke), B.2.3.2 (Intracerebral hemorrhage, ICH), B.2.3.3 (Subarachnoid hemorrhage, SAH), B.8.1.1 (Diabetes mellitus type 1, T1D), B.8.1.2 (Diabetes mellitus type 2, T2D), and B.8.2 (Chronic kidney disease, CKD). Three disease conditions, which are also comorbid factors of stroke, were also selected, high SBP (>110–115 mmHg), high BMI (>23.0 kg/m$^2$), and high LDL cholesterol. For these, ASMRs in 204 countries, as well as global rates for 2009–2019 were obtained using the GBD Results Tool, and age-standardized prevalence percentages in 204 countries for the years 2009–2016 (data for 2017–2019 was not available) were obtained using NCD-Risc website (ncdrisc.org/) (*Zhou et al., 2017*; *NCD Risk Factor Collaboration NCD-RisC, 2017*). Global crude rates and age-standardized incidence, prevalence, and mortality rates were obtained from GBD and NCD for all the diseases and disease conditions (*Figure 1—figure supplement 1*, *Supplementary file 1*). GBD 2019 and NCD-RisC study compiled with the GATHER Guidelines.

## Spatio-temporal trend analysis and estimated annual percent change

The 204 countries were grouped into eight geographic regions namely Global, America, Europe, Middle East, Africa, Central Asia and South Asia, East Asia, and Oceania, based on ethnicity (**Supplementary file 1**, **Figure 1—figure supplement 3**). The prevalence percentages were converted into ASPRs for each disease condition. ASMRs and ASPRs for each region were obtained using Bayesian model averaging of linear regression models with Markov chain Monte Carlo sampling. Bayesian Information Criterion (BIC) was the model selection criteria. Poisson distribution with global rates as lambda was the prior distribution on the models. The population of each country was used as weights. 10,000 draws from the posterior distribution of model parameters were used to obtain the point estimates (mean of the draws) and 95% uncertainty intervals (2.5th to 97.5th percentiles of the posterior distributions) of mortality and prevalence rates for each region for years 2009–2019. Rate estimates presented in this paper are age-standardized rates per 100,000 population.

The ASMRs and ASPRs thus obtained for the different diseases in each region were subjected to a temporal rank analysis for 2009, 2014, and 2019 using custom Perl scripts and change in the trend was plotted as a bump plot. The size and position of the points indicates the rate and rank, respectively. To quantify the temporal trends, EAPC was modeled using Poisson regression using a generalized linear model for the log-transformed rates: $\log(y) = \beta 0 + \beta 1 x 1 + \beta 2 x 2 + \ldots + \beta p x p$, where $y$ is the age-standardized rate, $x_i$ are the calendar years, $\beta_i$ are the rate trends. Under the assumption of linearity of log of age-standardized rate with time, EAPC $= 100 * \exp(\beta) - 1$. The 95% uncertainty interval is calculated as CI(EAPC) $= \beta + (Z_{(1-\alpha)/2}) \times$ SE, where $\alpha$ is the confidence level, SE is the standard error of $\beta$.

EAPC for ASMRs and ASPRs was calculated for the time period 2009–2019 (**Tables 1 and 2**). EAPC for high SBP prevalence was calculated from 2009–2015 and for high BMI prevalence from 2009 to 2016. EAPC was considered statistically significant if the uncertainty interval of EAPC did not cross zero. Statistical significance of spatio-temporal difference in rates was calculated using chi-square test by assuming the rate to be under Poisson distribution. 2019 ASMRs (or ASPRs) in each region were compared with global ASMR (or ASPR) to compare change over locations and with 2009 ASMRs (ASPRs) to compare change over time. 2014 ASPRs were used when 2019 data were not available. p-values (two-sided) for ASMRs and ASPRs are shown in **Supplementary file 1**. Chi-square tests were done using Open Source Epidemiologic Statistics for Public Health (https://www.openepi.com/Menu/OE_Menu.htm). The relation between ASMRs and ASPRs of each region was measured using Pearson correlation. For correlation, we considered the data of 2014 due to the complete spectrum of data availability. All analysis, unless specified, was done using R Statistical Software (version 4.1.2) (**R Development Core Team, 2021**) with packages **BAS** (**Clyde, 2022**), **Rcan** (**Laversanne and Vignat, 2020**), **corrplot** (**Wei and Simko, 2021**), **dplyr**, and **ggplot2**.

## Proportional mortality and prevalence

The diseases were classified into three categories as stroke (ischemic stroke, ICH, SAH, and IHD), metabolic disorders (high BMI, high LDL, T2D, CKD, and T1D) and high SBP. To determine the proportion of each category in a region, total ASMRs were scaled to 100 for all 3 years. The same was done for ASPRs (high LDL cholesterol data were not available). Statistical significance of difference in proportions was calculated using a one-sample test for binomial proportion using normal-theory method. The proportion mortality in each category was compared in a pairwise manner with global as well other regional proportions to calculate the two-sided p-value. The same was done for proportional prevalence. P-values for proportional mortality and prevalence are shown in **Supplementary file 1**. Proportion comparisons were done using Open Source Epidemiologic Statistics for Public Health (https://www.openepi.com/Menu/OE_Menu.htm).

## Population structure analysis

For evaluating the ethnogenetic perspective of stroke and its comorbid conditions we considered the risk variants associated with each disease. The risk variants were obtained from GWAS Catalog (https://www.ebi.ac.uk/gwas/home), during the period November 2021 to August 2022. The trait IDs used to retrieve data from GWAS Catalog for the different diseases were EFO_0000712 (stroke), EFO_0001645 (IHD), EFO_0000537 (high SBP), MONDO_0005148 (T2D), MONDO_0005147 (T1D), EFO_0003884 (CKD), EFO_0007041 (high BMI), and EFO_0004611 (high LDL). The position of the risk variants in GRCh37 assembly was obtained, and variants less than 10,000 bp apart were excluded

using custom Perl scripts. The total number of risk variants for each disease thus obtained are shown in *Supplementary file 1*. The genotype of identified biallelic autosomal SNPs in unrelated individuals was extracted from the 1000 Genome VCF files (https://www.internationalgenome.org/; *Lusis and Pajukanta, 2008*). The proportion of ancestral populations in each individual was estimated from their genotype using a model-based clustering approach (*Pritchard et al., 2000*). The admixture model with correlated allele frequencies was specified to cluster the individuals into either five or three clusters (in case of BMI). The genotype was converted to eigenvectors using principal component analysis (*Purcell and Chang, 2023*; *Chang et al., 2015*).

## Shared and unique risk variants among ethnicities

The number of GWAS risk variants for stroke shared among, as well as, unique to the five superpopulations in 1000 Genomes (African, East Asian, South Asian, European, and American) was obtained (*Figure 6* and *Supplementary file 1*). A risk variant was considered to be present in a population if the alternate allele frequency in 1000 Genomes was greater than or equal to 0.05. For each GWAS risk variant, the gene variant map was obtained, and the genes shared among the populations was estimated (*Supplementary file 1*).

## Calculation of LD of risk variants of stroke in South Asia

Proxy variants ($R^2 > 0.01$) in the region ±500 kb of the risk variant of stroke present in the South Asian population was obtained from LDlink (*Machiela and Chanock, 2015*). Among the proxy variants, variants with MAF less than 0.1 were selected along with the risk variant as low-frequency variants, and variants with MAF greater than 0.1 were termed as common variants. LD between two alleles A and B is quantified using the coefficient of LD $D_{AB}$ calculated using the equation $D_{AB} = p_{AB}p_Ap_B$, where $pi$ represents the frequency of the allele $i$ or haplotype $i$. To be able to compare the level of LD between different pairs of alleles, $D$ is normalized as follows: $D' = D/D\text{max}$, where $D\text{max} = \max\{-p_Ap_B, -(1\ p_A)(1 - p_B)\}$ when $D < 0$ and $D\text{max} = \min\{p_B(1\ p_A), p_A(1 - p_B)\}$ when $D > 0$ (*Lewontin, 1964*). Estimates of $D'$ were calculated separately for low-frequency variants and common variants, and plotted using the R library *gaston* (*Perdry and Dandine-Roulland, 2022*).

## Materials and correspondence

The datasets analyzed during the current study are available in the GBD database, at https://vizhub.healthdata.org/gbd-results/?params=gbd-api-2019-permalink/fe8b05e8222bcf3ec3af555762006f2a.

All other materials including computer code will be available upon request. Please contact the corresponding author for the same.

## Acknowledgements

RS acknowledges the support of Kerala State Council for Science, Technology and Environment (KSCSTE) for providing the research fellowship. RS and ASN acknowledge the SIUCEB support at the Department of Computational Biology and Bioinformatics, University of Kerala for providing the necessary facilities to carry out the work. MB acknowledges the Department of Biotechnology for providing intra-mural support to carry out the work.

## Additional information

### Funding

| Funder | Grant reference number | Author |
|---|---|---|
| Kerala State Council for Science, Technology and Environment | | Rashmi Sukumaran |

| Funder | Grant reference number | Author |
| --- | --- | --- |
| Rajiv Gandhi Centre for Biotechnology, Department of Biotechnology, Ministry of Science and Technology, India | | Moinak Banerjee |

The funders had no role in study design, data collection, and interpretation, or the decision to submit the work for publication.

## Author contributions

Rashmi Sukumaran, Resources, Data curation, Software, Formal analysis, Validation, Investigation, Visualization, Methodology, Writing – original draft; Achuthsankar S Nair, Conceptualization, Supervision, Investigation, Visualization, Project administration, Writing - review and editing; Moinak Banerjee, Conceptualization, Resources, Data curation, Software, Formal analysis, Supervision, Funding acquisition, Validation, Investigation, Visualization, Methodology, Writing – original draft, Project administration, Writing - review and editing

## Author ORCIDs

Rashmi Sukumaran ⓘD https://orcid.org/0000-0002-7951-3495
Moinak Banerjee ⓘD https://orcid.org/0000-0002-0842-0398

Reviewer #1 (Public Review): https://doi.org/10.7554/eLife.94088.3.sa1
Reviewer #2 (Public Review): https://doi.org/10.7554/eLife.94088.3.sa2
Author response https://doi.org/10.7554/eLife.94088.3.sa3

# Additional files

## Supplementary files

• Supplementary file 1. Supplementary information. (**a**) Global incidence, mortality, and prevalence rates for stroke, its subtypes, and comorbid factors. (**b**) Spatio-temporal comparison of mortality rates of stroke, its subtypes, and comorbid factors. (**c**) Spatio-temporal comparison of prevalence rates of stroke, its subtypes, and comorbid factors. (**d**) Comparison of proportional mortality of stroke, high SBP, and metabolic conditions. (**e**) Comparison of proportional prevalence of stroke, high SBP, and metabolic conditions. (**f**) Variants and genes associated with stroke and its comorbid factors that are shared among populations, as well as unique to populations. (**g**) Continent Regions. (**h**) Number of final risk variants for each disease.

• MDAR checklist

## Data availability

The datasets analyzed during the current study are available in the GBD database.

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
