## [Editor Report · eLife assessment]

This paper provides a **useful** analysis of the variation of the burden of strokes across geographic regions, finding differences in the relationship between strokes and their comorbidities. This dataset and the correlations found within will be a resource for directing the focus of future investigations. The results are technically **solid**, but there are cases where statistical analyses are yet to be carried out to support statements of statistical significance.

---

## [Referee Report · Reviewer #1 (Public Review)]

Summary:

The paper measures the prevalence and mortality of stroke and its comorbidities across geographic regions in order to find differences in risks that may lead to more effective guidance for these subpopulations. It also does a genetic analysis to look for variants that may drive these phenotypic variations.

Strengths:

The data provided here will provide a foundation for a lot of future research into the causes of the observed correlations as well as whether the observed differences in comorbidities across regions have clinically relevant effects on risk management.

The use of data from before COVID-19 is both a strength and a weakness. Because COVID had effects on vascular health and had higher death rates for groups with the comorbidities of interest here, it has likely shifted the demographics in ways that would shift the results in unpredictable ways if the analysis were repeated with current data. This can be a strength in providing a reference point for studying those changes as well as allowing researchers to study differences between regions without the complication of different public health responses adding extra variation to the data. On the other hand, it limits the usefulness of the data in research concerned with the current status of the various populations.

---

## [Referee Report · Reviewer #2 (Public Review)]

Summary:

The authors have analyzed ethnogeographic differences in the comorbidity factors, such as a diabetes and heart disease, for the incidences of stroke and whether it leads to mortality.

Strengths:

The idea is interesting and data are compelling. The results are technically solid when presented, but in many cases statistical analyses are yet to be carried out to support statements of statistical significance.

The authors identify specific genetic loci that increase the risk of a stroke and how they differ by region.

Weaknesses:

The presentation is not focused. It is important to include p-values for all comparisons and focus the presentation on the main effects from the dataset analysis.

---

## [Author Response]

The following is the authors’ response to the current reviews.

**eLife assessment:**

I find that the eLife assessment mentions “statistical analyses are yet to be carried out to support statements of statistical significance” while the reviewers mention that the data are compelling and results are technically solid. Besides all observations in the manuscript are presented with robust and established norms of statistical analysis.

**Public Reviews:**

**Reviewer #1 (Public Review):**
Strengths:The use of data from before COVID-19 is both a strength and a weakness. Because COVID had effects on vascular health and had higher death rates for groups with the comorbidities of interest here, it has likely shifted the demographics in ways that would shift the results in unpredictable ways if the analysis were repeated with current data. This can be a strength in providing a reference point for studying those changes as well as allowing researchers to study differences between regions without the complication of different public health responses adding extra variation to the data. On the other hand, it limits the usefulness of the data in research concerned with the current status of the various populations.

We completely agree with the observation, but were restricted as the purpose was to use the most robust and technically qualified data from GBD. The post COVID19 GBD data has not yet been released, but I am sure the observations made in the study can help in guiding the issues in the post COVID era too, because genetics is not going to change in these population groups.

However, we did highlight this aspect of COVID19 even in our original version and also in the revised version.

**Reviewer #2 (Public Review):**
Weaknesses:The presentation is not focused. It would be better to include p-values and focus presentation on the main effects from the dataset analysis.

The significant p-values were restricted to public health data only to identify and distinguish differences in incidence, prevalence and mortality and how they differ across world populations. These differences have often been interpreted from socio-economic point of view, while our manuscript presents the reasons for differences for main condition (Stroke) and its comorbid condition among different ethnicities from a genetic perspective. This genetic perspective was further explored to identify unique ethnic specific variants and their patterns of linkage disequilibrium in distinguishing the phenotypic variations. Considering the quantum and diversity of data, both for public health and GWAS data, there can be several directions but for presentation we focused only on the most distinguishing and established phenotypic differences. I am sure this will open up avenues for several future investigations including COVID, as has been highlighted by the reviewers too. All observations were presented with robust and established norms of statistical analysis.

The following is the authors’ response to the original reviews.

Thanks for the constructive observations on strengths and weaknesses of our manuscript. Interestingly, some of the weaknesses mentioned here also turns out to be the strength of the article. For example COVID19 has been mentioned by the reviewer as a driver to increase the mortality in some comorbid conditions and stroke. Firstly, I must clarify that, our data is from PreCOVID era and we indeed mention that in COVID era, COVID-19 might differentially impact the risk of stroke. Possibly this differential influence on the comorbidities of stroke, is likely to be influenced by its underlying genetics of stroke and its comorbidities.

I have tried to address the concerns raised by the reviewers, which ideally doesn’t impact the original manuscript. Statistical limitation has been commented pertaining to P-values, which has been clarified here. However, certain minor concerns such as abbreviations have been resolved in the revised manuscript. My response to weakness and reviewer’s comments are mentioned below.

**Reviewer #1 (Public Review):**
Strengths:The data provided here will provide a foundation for a lot of future research into the causes of the observed correlations as well as whether the observed differences in comorbidities across regions have clinically relevant effects on risk management.Weaknesses:As with any cross-national analysis of rates, the data is vulnerable to differences in classification and reporting across jurisdictions.

GBD data is the most robust and most comprehensive data resource which has been used and accepted globally in predicting the health metrics statistics.

GBD data indeed considers normalisations, regarding classification and reporting.

To the best of our knowledge this is the best available resource to consider all health metrics analysis.

Furthermore, given the increased death rate from COVID-19 associated with many of these comorbid conditions and the long-term effects of COVID-19 infection on vascular health, it is expected that many of the correlations observed in this dataset will shift along with the shifting health of the underlying populations.

I must clarify that we have used data prior to COVID-19.

But yes the patterns after COVID19 will shift due to the impact of covid. This makes the study even more relevant as the comorbid conditions of stroke are also the risk drivers for COVID19 and mortality. This shift has been reported by some authors, which has been discussed in the discussion.

Therefore, understanding the genetic factors underlying stroke and its comorbid conditions might help in resolving how COVID19 might differentially impact on health outcome.

We did highlight this aspect of COVID19 even in our original version.

Introduction 1st para:

“It is the accumulated risk of comorbid conditions that enhances the risk of stroke further. Are these comorbid conditions differentially impacted by socio-economic factors and ethnogeographic factors. This was clearly evident in COVID era, when COVID-19 differentially impacted the risk of stroke, possibly due to its differential influence on the comorbidities of stroke.”

Discussion 3rd para:

“Studies reported reduction in life expectancy in 31 of 37 high-income countries, deduced to be due to COVID-191 . However, it would be unfair to ignore the comorbid conditions which could also be the critical determinants for reduced life expectancy in these countries.”

**Recommendations For The Authors:**

On page 5, the authors make a note about Africa and the Middle East having the highest ASMR for high SBP and comment about the relative populations of these regions. The populations of the regions are irrelevant to the rate.

Since the study is on comorbid factors of stroke and its impact on mortality therefore, relative burden seems critical. This has been further elaborated here to justify the comment, which indeed is an integral part of the original manuscript.

Paragraph referred – Results section 2:

“Ethno-regional differences in mortality and prevalence of stroke and its major comorbid conditions

We observed interesting patterns of ASMRs of stroke, its subtypes and its major comorbidities across different regions over the years as shown in figure 1a, table 1 and supplementary file 1. When assessed in terms of ranks, high SBP is the most fatal condition followed by IHD in all regions, except Oceania (OCE) where IHD and high SBP swap ranks. Africa (AFR; 206.2/100000, 95%UI 177.4-234.2) and Middle East (MDE; 198.6/100000, 95%UI 162.8-234.4) have the highest ASMR for high SBP, even though they rank as only the third and sixth most populous continents (fig. S2), respectively.”

On page 17, the authors are alarmed by a large ratio between prevalence rates and mortality rates for certain conditions. This is confusing since this indicates that these conditions are not as dangerous as the other conditions.

This has been further elaborated here to justify the comment, which indeed is an integral part of the original manuscript.

Paragraph referred – Discussion para 1:

“While the global stroke prevalence is nearly 15 times its mortality rate, prevalence of comorbid conditions such as high SBP, high BMI, CKD, T2D are alarmingly 150- to 500-fold higher than their mortality rates. These comorbid conditions can drastically affect the outcome of stroke.”

In Figure 4, the colors are not defined.

In Structure plot colours are assigned as per each K, it doesn’t directly refer to any population. But the plot distinguishes the stratification of populations as per K. Ramasamy, R.K., Ramasamy, S., Bindroo, B.B. et al. STRUCTURE PLOT: a program for drawing elegant STRUCTURE bar plots in user friendly interface. SpringerPlus 3, 431 (2014). https://doi.org/10.1186/2193-1801-3-431

**Reviewer #2 (Public Review):**
Strengths:The idea is interesting and the data are compelling. The results are technically solid.The authors identify specific genetic loci that increase the risk of a stroke and how they differ by region.Weaknesses:The presentation is not focused. It would be better to include p-values and focus presentation on the main effects of the dataset analysis.

I presume the comment is made with reference to results with significant p-values.

P-values are mentioned in the main text when referring to significant decrease or increase with respect to global rates and time e.g. P-values for comparison of a year 2019, are based on regional rates to global rates of 2019 in Supplementary file 1b (mortality) and 1c (prevalence) e.g. P-values for comparison between year is based on 2019 rates to 2009 rates in Supplementary 1b (mortality) and 1c (prevalence) e.g. P-values for proportional mortality and proportional prevalence in Supplementary file 1d and 1e is also based on global rates.

**Recommendations For The Authors:**
It would be better to minimize the use of acronyms. Often one has to go back to decipher what the acronym stands for. It is fine to have acronyms in figure legends, if necessary. However, at least for regions, please do not use acronyms.

In the revised version we have tried to minimise the Acronyms.

Removed the acronyms for regions and other places wherever possible however, the diseases acronyms have been maintained as per the GBD terms.

Please focus the presentation on the main results. Currently, the presentation wanders and repeats itself a lot.

Since the manuscript tries to address the global and regional rates of prevalence, mortality and its relationship to genetic correlates, it is difficult not to repeat the same to stress the significant observations coming out of different analysis methods. This might reflect on some amount of repetitiveness but the intention was to stress the significant observations.

I would also recommend acknowledging and discussing socioeconomic factors earlier in the manuscript.

Current mention happens in 3rd para of Discussion

“The changing dynamics of stroke or its comorbid conditions can be attributed to multitude of factors. Often global burden of stroke has been discussed from the point of view of socio-economic parameters. Studies indicate that half of the stroke-related deaths are attributable to poor management of modifiable risk factors 8,9. However, we observe that different socio-economic regions are driven by different risk factors.”